# Procedural Image Programs for Representation Learning

**Manel Baradad**[1], **Chun-Fu (Richard) Chen**[2], **Jonas Wulff**[3], **Tongzhou Wang**[1],
**Rogerio Feris**[4], **Antonio Torralba**[1], **Phillip Isola**[1]
[1]MIT CSAIL, [2]JPMorgan Chase Bank, N.A., [3]Xyla, Inc, [4]MIT-IBM Watson AI Lab

## Abstract

Learning image representations using synthetic data allows training neural networks
without some of the concerns associated with real images, such as privacy and
bias. Existing work focuses on a handful of curated generative processes which
require expert knowledge to design, making it hard to scale up. To overcome
this, we propose training with a large dataset of twenty-one thousand programs,
each one generating a diverse set of synthetic images. These programs are short
code snippets, which are easy to modify and fast to execute using OpenGL. The
proposed dataset can be used for both supervised and unsupervised representation
learning and reduces the gap between pre-training with real and procedurally
generated images by 38%. Code, models, and datasets are available at: `https:
//github.com/mbaradad/shaders21k`

## 1 Introduction

Training neural networks using data from hand-crafted generative models is a novel approach that
allows pre-training without access to real data. This technique has been shown to be effective on
downstream tasks for different modalities, such as images [1, 2], videos [3], and text [4]. Despite its
big potential, previous work for images focuses on a handful of generative processes, like fractals [5],
textured polygons [3], or dead-leaves and statistical processes [2].

To achieve good performance with the original set of procedures, expert knowledge is required to
make the generated images match simple image statistics. This makes it hard to improve over existing
work, as these generative models have already been carefully studied and tuned. What would happen
if we used a large set of procedural image programs without curation, instead of focusing on a small
set of programs?

To this end, we collect a large-scale set of programs, each producing a diverse set of images that
display simple shapes and textures (see Figures 1 and 2 for examples). We then use these programs to
generate images and train neural networks with supervised and unsupervised representation learning
methods. These networks can then be used for different downstream tasks, reaching state-of-the-art
performance for methods that do not pre-train using real images.

The generative programs of this collection are coded in a common language, the OpenGL shading
language, that encapsulates the image generation process in a few lines of code. Previous work uses
ad-hoc slow methods for each generative process, which makes it hard to integrate them into the
same framework. Compared to existing approaches, the rendering of our programs is performed at
high throughput (hundreds of frames per second with a single modern GPU), allowing generation on
the fly while training.

As the dataset consists of a large collection of generative processes, we can use each code snippet as a
latent class, which allows learning representations using supervised approaches. In contrast, previous
methods require ad-hoc techniques for clustering the generative parameters into discrete classes [5].

36th Conference on Neural Information Processing Systems (NeurIPS 2022).

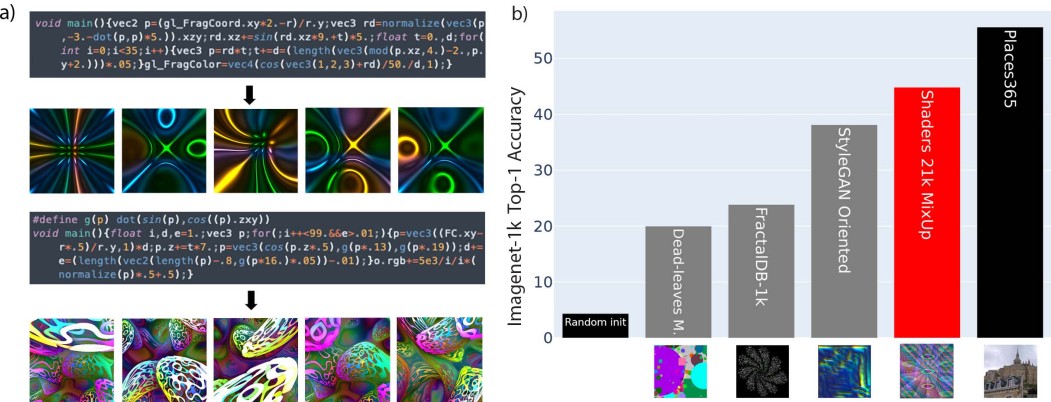

Figure 1: a) We propose learning image representations using a vast collection (21k) of short OpenGL fragment shaders. The code and rendered images for two of them are shown. b) Top-1 Accuracy on ImageNet-1k for linear classification with a ResNet-50 pre-trained with procedurally generated images from previous methods (Dead-Leaves Mixed [2], FractalDB [1] and StyleGAN Oriented [2], in gray) and our best generative image model, Shaders-21k with MixUp (in red). The lower and upper-bound performance (in black) is for a random initialization and pretraining with Places365 [6].

Together with the collection of programs, the main contributions of the paper are:

- A study on how performance scales as a function of the number of generative image programs, for supervised and unsupervised approaches. Our experiments show that downstream performance scales logarithmically with the number of generative programs (see Figure 3 and Section 4).
- State-of-the-art performance for representation learning without real data. When the number of programs is high enough, our approach outperforms previous methods by a large margin.
- An analysis of the properties that make for a good image generation program in Section 5. This provides insights on how to design new programs that maximize performance.

Previous work [2] showed that performance saturates fast when generating more samples from their data distribution (i.e. more images from their curated set of programs), while our work shows that more samples from our distribution (i.e. more programs) do not saturate performance at the current scale. This points out a clear direction on how to further improve performance: collect or learn to generate more image programs.

To the best of our knowledge, this is the first work to pair a large collection of image-generation processes and short programs that is amenable to experimentation. This set of procedural programs has the potential to be a testbed to answer key questions linked to these paired modalities, some of which we answer in this paper.

## 2    Previous work

**Pre-training with synthetic data** Pre-training neural networks is a well-known technique to improve performance and convergence speed during training. Typically, pre-trained networks are either used as a starting point in the optimization or to transfer the knowledge from related tasks to the one of interest [7].

When available, pre-training uses real data as close as possible to the task of interest. If the data available for the task is scarce, a common practice is to train on a dataset of natural images, typically ImageNet [8]. These pre-trained models are then used on a wide variety of tasks, achieving state-of-the-art results on tasks with different degrees of similarity to ImageNet. Despite this, it is unclear how much of the performance gain is due to knowledge transfer from the pre-training task or the optimization procedure being simpler to engineer when starting from a pre-trained model [9].

Alternatively, other works achieve state-of-the-art performance by pre-training on synthetic data. These approaches generate realistic-looking images paired with ground-truth annotations, which are then used as pre-training data. Currently, this approach is a standard technique to obtain state-of-the-

art performance for low-level vision tasks such as flow estimation [10, 11] and depth prediction [12], between others.

Contrary to these approaches, recent works have studied whether it is necessary to use real-looking data for pre-training. The motivations for doing so are several folds, from targeting domains where rendering would require expert knowledge (like medical imaging), deploying agnostic models to downstream tasks, or making the generative process interpretable and optimizable [3].

In this spirit, the work of [1] introduced a way to train image models using renders with fractal structures. This procedure has been shown to work with different neural architectures [5] and has been further improved with better sampling and colorization strategies [13]. In the same spirit, [2] tested different generative processes to learn general image representations. Their best-performing models are based on neural networks, which are not interpretable and require expert knowledge to design manually.

**Representation learning** Representation learning is at a level of maturity that shallow networks on top of pre-trained models outperform supervised approaches trained from scratch [14, 15]. Training on a large corpus with supervision is currently the best representation of learning practice in terms of performance [16].

On the other hand, unsupervised approaches allow training with unlabeled data, which is usually easier to acquire than labeled data. Although both outperform models trained with smaller data collections, this comes at the price of more severe ethical issues, as models inherit the concerns associated with the datasets they are trained on.

The most widely used method for unsupervised representation learning is contrastive learning [17], which has seen consistent improvements in recent years [14, 18]. Contrastive learning is well established and has been shown to perform well in a wide range of image domains, including medical imaging [19] and satellite imagery [20]. Because of this, in this work we focus on contrastive learning as an unsupervised approach to learning image representations.

Alternatively, early work on unsupervised representation learning focused on pretext tasks that are different from contrastive training, such as context prediction [21] or rotation prediction [22]. Although these did not achieve the levels of performance of contrastive learning, recent work has shown that one of these pretext tasks, masked autoencoding with an image prediction loss, coupled with Vision Transformers [23], can outperform contrastive training in large-scale settings [24]. We do not study these novel approaches as their robustness and broad applicability to different tasks and small training budgets is yet to be studied.

## 3 The Shaders-1k/21k dataset

The collection of programs we propose consists of a set of generative processes of the form: $g_\theta^i : \mathbf{z} \to \mathbf{x}$. Each generative program corresponds to the latent class $g_\theta^i$, from which we can produce images $x$ by sampling the stochastic variables $z$ from a prior model. The constant $i$ indexes the programs available, while $\theta$ corresponds to the numerical constants of the generative program. In the proposed collection, each $g_\theta^i$ corresponds to an OpenGL fragment shader. Existing similar methods [1, 2] focus on a single type of program ($i = 1$, with variable $\theta$), while this work focuses on a large collection of generative procedures ($i \in \{1, ..., 21k\}$) with fixed $\theta$.

In previous work, the generative programs were designed ad hoc in different programming languages, making it hard to scale or mix different approaches. To overcome this, we fix a common programming language and choose the OpenGL Shading Language, a high-level programming language that allows specifying rendering programs with a syntax based on the C programming language. Furthermore, it is amenable to GPU acceleration, making it particularly suitable to quickly produce images using a standard deep learning stack.

### 3.1 Program collection

To obtain the desired OpenGL fragment shaders, we query the web for code snippets corresponding to fragment shaders. We obtain them from two sources: Twitter and Shadertoy [25]. The programs obtained from Twitter use the TwiGL syntax as described in [26], which allows for extremely compact code (typically less than 280 characters) by abstracting away standard functionalities. On the other

| Property | | Avg. | $Q_5$ | $Q_{95}$ |
|---|---|---|---|---|
| # chars | T | 283.8 | 160 | 662 |
| | S | 4307.2 | 499 | $13.6k$ |
| JPEG (KB) | T | 70.1 | 15.0 | 149.5 |
| | S | 53.1 | 8.8 | 142.0 |
| gzip (KB/img) | T | 57.5 | 9.8 | 124.0 |
| | S | 42.0 | 3.6 | 117.4 |
| FPS | T | 886.9 | 331.0 | $1.3k$ |
| | S | 984.4 | 293.5 | $1.4k$ |

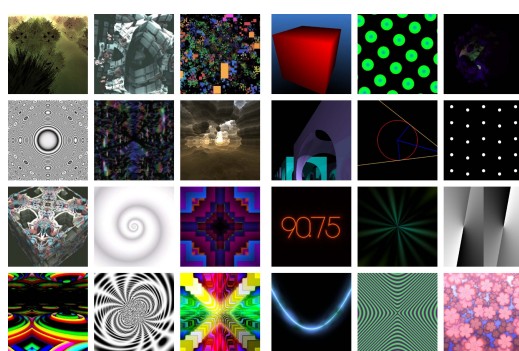

Table 1: Dataset properties across shaders for TwiGL (T) and Shadertoy (S), reported as average and 5/95-quantiles over all shaders of each subset. Statistics per shader have been computed with 400 samples at a resolution of $384 \times 384$, and rendering time is computed using a single Nvidia GeForce GTX TITAN X, including transfer to general memory.

Figure 2: Example images for a single random timestep $t$ for several random shaders. The first three columns correspond to shaders from TwiGL (Shaders-1k), while the last three columns are shaders from Shadertoy (only included on Shaders-21k). As it is apparent, the complexity, of TwiGL shaders tends to be superior.

hand, Shadertoy is a repository of OpenGL shaders using the standard OpenGL syntax. Shaders in Shadertoy are generally longer (on average 4k characters) and they produce simpler images than those from TwiGL, as the latter are typically coded by more expert users.

The only stochastic variable we consider in practice in the collected shaders is the timestep ($\mathbf{z} = t \in \mathbb{R}$). The parameter $t$ allows generating a continuous video stream, which in lots of cases is periodic. We sample $t$ at 4 frames per second for as many samples as required through all experiments. We add an increment to $t$ sampled at uniform $\Delta t \sim \mathcal{U}[0, 0.25]$, to avoid exact duplicates for the cases where the shader is periodic.

After downloading the code snippets from these two sources, we obtain an initial set of $52k$ shaders that compile with the OpenGL shading language syntax. We first perform duplicate removal by rendering a single image using the same $t$ for all scripts and removing shaders that produce the same images. Additionally, we remove scripts for which different $t$'s produce the same image (i.e. they produce static images). With this, we obtain a final set of 1.089 fragment shaders from TwiGL and 19.994 from Shadertoy, which are unique and generate a diverse set of images when varying $t$.

In Table 1, we report the main characteristics of the shaders and the images they produce. Combining all the shaders from both sources, images can be rendered at 979 frames per second at a resolution of $384 \times 384$, using a single modern GPU and including transfer to general memory. When stored to disk as JPEG, the average size per image is 54 kB, compared to 70 kB for ImageNet-1k when resized to the same resolution.

**Shaders-1k/21k** From a qualitative analysis and the compression results of the generated images in Table 1, we further conclude that TwiGL shaders are more curated and of higher quality than those obtained from Shadertoy (see Figure 2 for examples of both). To distinguish and experiment with both sources, we refer to the dataset composed of only TwiGL shaders as Shaders-1k and the dataset consisting of all available shaders from both sources as Shaders-21k.

## 4 Experiments

### 4.1 Supervised vs Unsupervised representation learning

To test downstream performance when pre-training with Shaders-1k and Shaders-21k, we perform an initial set of experiments with and without latent class supervision. We test three representation learning methodologies: supervised classification with cross-entropy loss (CE), supervised contrastive learning (SupCon) [27], and unsupervised representation learning (SimCLR) [14].

| Dataset | S.CLR | CE | S.Con |
|---|---|---|---|
| Random init. | 17.10 | 17.10 | 17.10 |
| Places | 52.20 | 50.7 | 55.22 |
| I-100 | 60.30 | 78.2 | 80.18 |
| S.GAN O. | 44.60 | - | - |
| D.leaves M. | 30.56 | - | - |
| FracDB-1k | 33.12 | 26.5 | 32.98 |
| S-1k | 43.26 | 41.2 | 40.54 |
| S-21k | 47.74 | 40.3 | **48.10** |

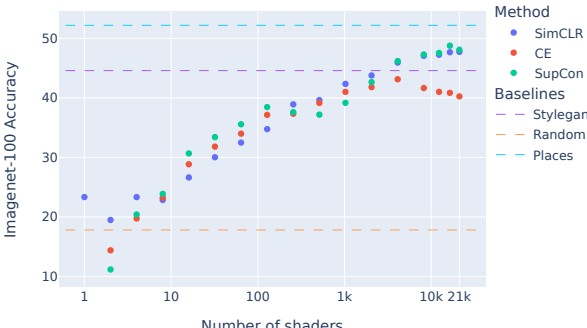

Table 2: Top-1 accuracy with linear evaluation on ImageNet-100, for a ResNet-18 pre-trained with supervised methods (CE and SupCon) and an unsupervised method (SimCLR), with 100k examples and all the classes (if available). Underlined results are upper bounds for real images.

Figure 3: Linear classification performance on ImageNet-100 using a ResNet-18 trained with three representation learning methods and 100k images while varying the number of shaders. As dashed lines, we show performance with SimCLR trained on StyleGAN Oriented (state-of-the-art baseline), Places365 (upper bound), and a random network (lower bound).

To compare these training methodologies, we train a ResNet-18 with 100k samples generated for each dataset at a resolution of $256 \times 256$. We follow the training procedure and public implementation described in [27] for the three methods. We train for 200 epochs with a batch size of $256$, with the rest of the hyperparameters set to those found to work well for ImageNet-100 in the original paper. The images after augmentations are fed to the network at a resolution of $64 \times 64$. After training the encoder, we evaluate using the linear protocol described in [17], which consists of only training a linear classifier on top of the representations learned. We evaluate on ImageNet-100, a subset of ImageNet-1k [8] defined in [28] using the averaged pooled features from the last convolutional layer, which have a dimensionality of 512.

As shown in Table 2, we compare performance using the shader datasets against Places365 [6], which was proposed in [2] as an upper bound for real images when testing on ImageNet; two procedurally generated datasets, FractalDB [1] and Dead Leaves with mixed shapes [2]; and the state of the art for training with similar data, StyleGAN Oriented [2].

Shaders-21k using SupCon achieves the best performance over existing methods that do not use real data, but it is closely followed by the same dataset when using SimCLR. Both methods performing similarly is expected in this case, as the number of class positives per batch decreases with the number of classes in the dataset. In this experiment, the batch size is relatively small (256) compared to the number of classes (21k). As the number of classes increases, the SupCon loss tends to be equivalent to SimCLR loss, and in the limit of having as many classes as training samples, they are the same.

Using this setting, we test performance for an increasing number of generative programs, sampled at random from Shaders-21k. In this experiment, the number of images is kept constant at 100k while the number of shaders increases (i.e. with more shaders, there are fewer images per shader). As seen in Figure 3, we observe that supervision is helpful in the middle regime where there are more than 20 shaders but less than 1000 shaders, but fails to outperform unsupervised approaches outside this regime. We also note from this sweep that performance is not saturated at 21k shaders, and it scales logarithmically with the number of shaders.

Finally, we also note from this sweep that using the full set of Shaders-1k outperforms a random subset of Shaders-21k of the same size. Shaders-1k achieves higher top-1 accuracy on ImageNet-100 (43.26) compared to a random subset of $1089$ shaders obtained from Shaders-21k (42.26). This further verifies that the shaders of Shaders-1k are on average of higher quality than those on Shaders-21k, as hypothesized in Section 3.

| Pre-train Dataset | I-1k | I-100 | VTAB Nat. | VTAB Spec. | VTAB Struct. |
|---|---|---|---|---|---|
| Random init | 4.36 | 10.84 | 10.98 | 54.30 | 22.64 |
| Places365 [6] | 55.59 | 76.00 | 59.72 | 84.19 | 33.58 |
| ImageNet-1k | 67.50 | 86.12 | 65.90 | 85.02 | 35.59 |
| StyleGAN O. [2] | 38.12 | 58.70 | 54.19 | 81.70 | **35.03** |
| StyleGAN O. [2] (MixUp) | 31.73 | 53.44 | 51.26 | 81.39 | 33.21 |
| FractalDB-1k [1] | 23.86 | 44.06 | 38.80 | 76.93 | 31.01 |
| Dead-leaves Mixed [2] | 20.00 | 38.34 | 35.87 | 74.22 | 30.81 |
| S-1k | 16.67 | 34.56 | 32.39 | 75.28 | 28.23 |
| S-1k MixUp | 38.42 | 60.04 | 53.24 | 82.08 | 30.32 |
| S-21k | 30.25 | 51.52 | 45.23 | 80.75 | 32.85 |
| S-21k MixUp | **44.83** | **66.36** | **57.18** | **84.08** | 31.84 |

Table 3: Top-1 accuracy with linear evaluation on ImageNet-1k/100 and the VTAB suite of datasets (averaged over the Natural, Specialized, and Structured categories), for a ResNet-50 trained with MoCo V2 with 1.3M samples for each of the pre-training datasets. Underlined results correspond to an upper bound (training with natural images different than the evaluation distribution) and the previous state-of-the-art without real data StyleGAN Oriented [2].

## 4.2 Large scale unsupervised training

Following the observation that contrastive training performs better than classification to pre-train feature extractors, we extend the previous approach to a large-scale setting. In this setting, we train a ResNet-50 using MoCo v2 [18], with images generated at $384 \times 384$ resolution. We train for 200 epochs with 1.3M images, with a batch size of 256, and set the rest of the hyperparameters to those found to work best on ImageNet-1k in the original paper. In Table 3, we show downstream performance for ImageNet-1k and ImageNet-100 and the 19 datasets of the Visual Task Adaptation Benchmark [29], averaged over the natural, specialized and structured categories.

When using data directly rendered from Shaders-1/21k, the performance is low compared to existing methods. We observe that the contrastive loss is lower for Shaders-1k compared to ImageNet-1k (6.2 and 6.6), while top-1 accuracy for the contrastive task is much worse for Shaders-1k (12.5% and 92.2% respectively). This points out that the network is able to solve the contrastive task properly on average (i.e. it is highly confident on most of the 64k negatives in the MoCo buffer), while it is highly confused by a few of the negative samples, which correspond to images from the same shader.

We hypothesize that this is caused by a large capacity network being able to learn shortcut solutions for the contrastive task [30], which results in learned features that do not transfer to downstream tasks. As can be seen from examples in Figure 2, it is easy for a large capacity network to distinguish different shaders via shortcut solutions, like background color or salient features, even after MoCo v2 image augmentations.

To overcome this, we propose mixing samples from different shaders in order to produce interpolated samples in pixel space. This removes some of the shortcut solutions while preserving the complexity of the underlying processes being mixed. We investigate three well-known mixing strategies: CutMix [31], MixUp [32], and producing samples from a GAN [33] that has been trained on the shaders. MixUp has been previously studied in the context of unsupervised contrastive learning [34], showing improved performance in the case where the train time distribution is aligned with the test time distribution.

Using FID computed with 50k samples with respect to ImageNet-100, we conclude that MixUp outperforms the other two strategies (details in Supp. Mat.). This metric was found to correlate with downstream performance in [2], and allows searching for good hyperparameters without training. For the MixUp strategy, we found that mixing 6 frames with weights sampled from a Dirichlet distribution with $\alpha_i = 1$ yields the best FID, while rendering time is still affordable. For a fair comparison against previous methods, MixUp is performed offline before training starts, so that the amount of images seen during training time is the same for all datasets (i.e. 1.3M fixed images, with the same MoCo v2 augmentations applied during training across all experiments).

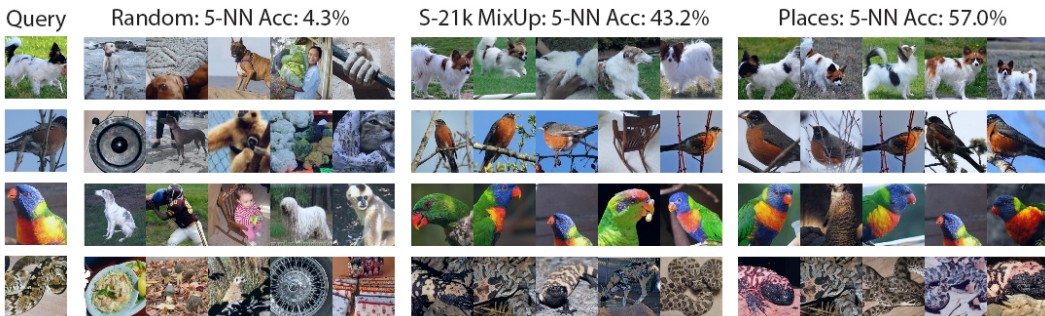

Figure 4: 5 nearest neighbors on ImageNet-100 for ResNet-50 randomly initialized (left) or trained with MoCo v2 on Shaders-21k with MixUp (middle) and Places365 (right). Reported accuracy corresponds to 5-NN accuracy on ImageNet-100.

As seen in Table 3, training with MixUp greatly outperforms the raw shaders. Shaders-21k with MixUp achieves the best performance overall on ImageNet-1k/100 and outperforms the previous state-of-the-art by a substantial margin: an extra 6.71% top-1 accuracy on ImageNet-1k. This represents a 38% relative improvement for the gap between real images different than ImageNet-1k (Places365), and the previous state-of-the-art for procedural images, StyleGAN Oriented [2]. In Figure 4 we show 5 nearest-neighbors retrieval on Imagenet-100 for our best model against a random initialized network and the Places365 network (additional results for other networks can be found in Supp.Mat.). As can be seen, nearest neighbors successfully retrieves good candidates given a query, capturing different perceptual aspects of the image that are not captured with a randomly initialized model.

We note that StyleGAN Oriented [2] does not improve with the MixUp strategy, achieving worse downstream performance than with the original samples. This is consistent with the FID metric, as the samples with MixUP yield worse (higher) FID than the samples without (38.74 vs 37.74).

In the case of VTAB, Shaders-1k/21k with MixUp outperforms the baselines for natural and specialized tasks, in the case of specialized tasks being close to Places performance. On the other hand, Shaders-1k/21k without MixUp outperform their MixUp counterparts on structured tasks, although performance for structured tasks is generally low for methods trained with contrastive learning. This is because structured tasks require information about counts, position, or size of the elements in the images, and this information is lost given the invariances that the data augmentations impose [35]. From the results per dataset (see Supp. Mat.), it can be seen that Shaders-1k performs well (even better than ImageNet) on datasets that resemble shaders, as is the case of the dSprites dataset, which consists of 2D shapes procedurally generated.

Finally, we use the fact that the Shaders dataset allows rendering novel samples during train time to train with the same procedure, but with a live generator (i.e. the samples for each of the 200 epochs are different). This methodology requires twice as much time to train with the same compute power, as the GPU is shared by the rendering and the training. Despite this, it only improves performance marginally, achieving a top-1 accuracy on ImageNet-1k/100 of 45.24% and 66.42%, respectively. This shows that, similarly to other works, scaling up the number of samples per shader (and not the number of shaders) has limited benefits, as this corresponds to 200 times more samples (corresponding to the 200 epochs) than the results presented in Table 3.

## 5 What makes for a good generative image program?

**A single shader:** To test what properties of a single shader are useful for downstream performance, we train a ResNet-18 with SimCLR as described in Section 4.1 for each shader in Shaders-1k. We use 10k images from each of the programs in Shaders-1k, train the ResNet-18 for 100 epochs, and then compute performance with linear evaluation for 100 epochs on a random subset of 10k images from ImageNet-100. As shown in Figure 5, there is substantial variability in performance between each of the shaders, and one of the main qualitative drivers for performance is the diversity between samples.

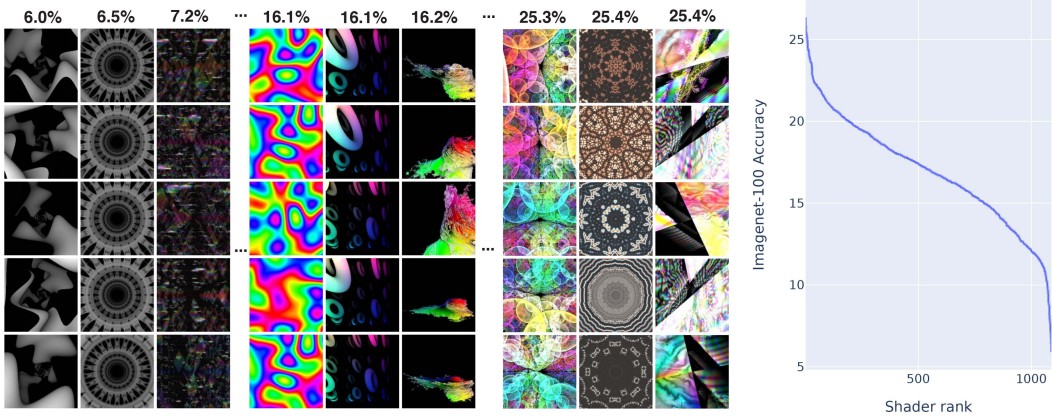

Figure 5: **Left:** Performance for the shaders in Shaders-1k trained with SimCLR on 10k images, evaluated on 10k samples of ImageNet-100. Left-to-right: images for different shaders sorted by performance (on top of each block). **Right:** I-100 accuracy of shaders in Shaders-1k, sorted by rank.

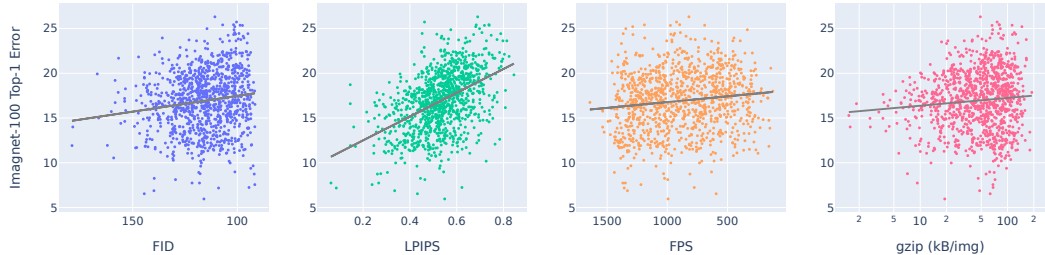

Figure 6: I-100 Performance against FID, LPIPS self-similarity, rendering time (frames per second), and gzip compression (for 400 images). Performance increases with lower FID (activations more similar to ImageNet images), lower self-similarity, higher rendering time (low FPS), and higher gzip file size, though the correlation is weak except for LPIPS self-similarity.

With the performance computed for each of the shaders in Shaders-1k, we train two shallow networks, consisting of two linear layers with ReLu and a hidden dimension of 32 to predict performance per shader. The inputs to the shallow network are the features extracted with the ResNet-50 trained with MoCo v2 for Shaders-1k with MixUp in Section 4.2. We train using an L1-loss to regress top-1 accuracy for each shader, splitting them 85-15 into train-val. Of the two trained networks, one gets information from 50 frames (as the concatenated average, minimum, and maximum of the activations) while the other only takes a single image as input. We use these networks for the rest of the experiments in this section, depending on whether the phenomena studied requires predicting performance from a single or multiple frames.

With the network that gets information from 50 frames, we rank all remaining shaders in Shaders-21k. In Figure 7 we show the shader with the best predicted performance over all Shaders-21k, which is more qualitatively diverse than the top-performing in Shaders-1k (see in Figure 5)and its top-1 empirical accuracy is 36%, considerably higher than the 26% achieved for the best shader in Shaders-1k.

Additionally, in Figure 6 we show scatter plots for several simple properties of the generated images for each shader against their performance. As can be seen, the metric that best correlates with performance ($r = 0.46$) is LPIPS intra-image distance, matching the findings in [2]. LPIPS [36] is a perceptual similarity metric that has been found to align better with human judgment than other alternatives. LPIPS intra-image is defined as the average LPIPS distance for two random crops covering $50\%$ of a given image and was first proposed in [37]. Consequently, we conclude that shaders that produce images with high self-similarity tend to perform poorly.

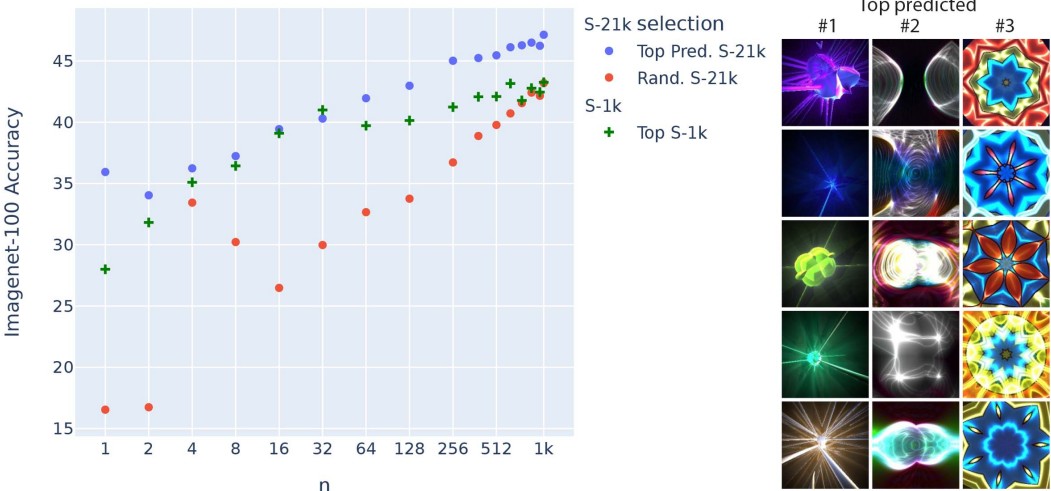

Figure 7: I-100 Top-1 accuracy with 100k images and SimCLR with different sets of shaders selected: 1) at random from Shaders-21k and 2) by predicted performance from all Shaders-21k that remain unseen while training the predictor. As a reference, we also plot the performance of the top shaders of Shaders-1k, but we note that this is a more limited set than the full Shaders-21k. The right-most columns are images from the top predicted shaders from Shaders-21k, the best best-performing one achieving 36% top-1 accuracy empirically, compared to 26% for the best in Shaders-1k.

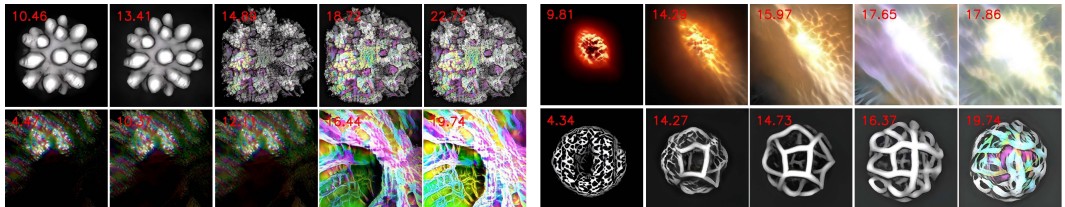

Figure 8: Walks on $z$ space for a StyleGAN trained with Shaders-1k, starting at a random $z$ (center column of each block) and following the trajectory that minimizes (first two columns) or maximizes (last two columns) performance. Factors that drive the performance include color diversity, filling the whole canvas, and intra-diversity, following the findings of the metrics in Figure 6.

On the other hand, although FID and gzip compression have a low correlation with performance, the plots show that there is a Pareto frontier for both metrics, where shaders cannot achieve high accuracy with low compression or high FID. Performance as a function of rendering time also suggests that there is a big margin for improvement, as the correlation between rendering time and performance is weak and there are lots of shaders that perform well while their rendering time is low.

**A collection of shaders:** In the same spirit, we consider what makes a collection of shaders perform well or bad when treated as a set, and whether it is possible to achieve increasingly better performance with a fixed number of shader programs. In Figure 7 we compare using a variable number of shaders selected either by predicted performance (with the same predictor as before that takes 50 frames) or at random. As seen in Figure 7, the greedy approach of each time selecting the shader with the best-predicted performance outperforms by a big margin random selection.

Finally, as a qualitative experiment of the properties of the Shaders-1k dataset that make for a good shader, we explore the space of a StyleGAN v2 trained with Shaders-1k, with the shallow predictor that only takes features from a single image. We explore StyleGAN space by tracing walks in the embedding space $z$, in the direction that minimizes or maximizes performance. As Figure 8 shows, samples that improve performance are more colorful and fill the canvas. On the other hand, samples obtained after traversing $z$ in the direction that minimizes performance tend to collapse to repetitive shapes, gray-scale images, or big regions of the image of solid color.

# 6 Conclusions

This paper proposes using a large collection of procedural image programs to image representations without real images. As experiments show, training with shaders outperforms existing methods that do not use real data. Using these procedural image models is competitive against natural images, especially in domains far away from the natural image domain, as demonstrated in the specialized tasks in VTAB.

## Acknowledgement

Manel Baradad was supported by the LaCaixa Fellowship, and this research was supported by a grant from the MIT-IBM Watson AI lab. Rogerio Feris was supported in part by Darpa LwLL. Experiments were partially conducted using computation resources from the Satori cluster donated by IBM to MIT, and the MIT's Supercloud cluster.

## Disclaimer

This paper was prepared for information purposes by the teams of researchers from the various institutions identified above, including the Global Technology Applied Research group of JPMorgan Chase Bank, N.A.. This paper is not a product of the Research Department of JPMorgan Chase Bank, N.A. or its affiliates. Neither JPMorgan Chase Bank, N.A. nor any of its affiliates make any explicit or implied representation or warranty and none of them accept any liability in connection with this paper, including, but limited to, the completeness, accuracy, reliability of information contained herein and the potential legal, compliance, tax or accounting effects thereof. This document is not intended as investment research or investment advice, or a recommendation, offer or solicitation for the purchase or sale of any security, financial instrument, financial product or service, or to be used in any way for evaluating the merits of participating in any transaction.

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
