# Procedural Image Programs for Representation Learning - Supplementary Material

## Contents

36th Conference on Neural Information Processing Systems (NeurIPS 2022).

# S-1    Analysis of mixing approaches

Training with the images directly sampled from shader programs performs badly on MoCo v2, as described in the main paper. Because of this, we test MixUp and CutMix to produce interpolated samples, with the purpose of avoiding shortcut solutions for the contrastive task. Due to computational constraints, we are not able to evaluate the full hyperparameter space of these three mixing strategies by training MoCo v2 for all possible configurations. We use FID as the metric for fast hyperparameter search, as it has been shown to correlate with downstream performance [1], and validate the results by training MoCo v2 with the best hyperparameters found with FID, as they appear in Section 4 of the main paper. In Table 1, we report the results for several mixing strategies for the datasets on which we experimented in Section 4.

To find the best hyperparameters for these interpolation strategies, we compute FID against Imagenet-100 with 50k images for both datasets. Using this criterion, we found that the gain in FID for the state-of-the-art synthetic dataset (StyleGAN Oriented) is marginal, while Shaders-21k benefits substantially from the mixing process. For Shaders-21k, we found that using between 4 and 8 samples performs similarly well, and we fixed the mixing strategy to 6 samples, as it yields a good balance between FID, sample diversity, and rendering time.

| Mixing Method | N | Places | I-100 | StyleGAN O. | Fractals | Dead-leaves M. | S-21k |
|---|---|---|---|---|---|---|---|
| None | 1 | 33.92 | 0.00 | 37.74 | 52.85 | 46.09 | 41.00 |
| MixUp | 2 | **32.70** | **5.44** | 37.86 | 46.63 | 45.22 | 37.45 |
| | 3 | 33.22 | 9.27 | 38.10 | 45.21 | 45.63 | 35.66 |
| | 4 | 33.89 | 12.57 | 38.31 | 44.90 | 45.71 | 35.08 |
| | 5 | 34.53 | 15.54 | 38.55 | 44.78 | 45.40 | **35.03** |
| | 6 | 35.10 | 18.01 | 38.74 | 44.85 | 45.09 | 35.04 |
| | 7 | 35.71 | 19.93 | 38.93 | 45.00 | 44.76 | 35.06 |
| | 8 | 35.98 | 21.78 | 39.05 | 45.09 | 44.40 | 35.05 |
| | 9 | 36.43 | 23.21 | 39.23 | 45.23 | 44.26 | 35.21 |
| CutMix | 2 | 32.89 | 6.46 | **36.97** | 46.79 | 44.50 | 36.09 |
| | 3 | 33.71 | 10.77 | 37.09 | 45.56 | **44.15** | 35.47 |
| | 4 | 34.15 | 13.09 | 37.41 | 45.08 | 44.08 | 35.44 |
| | 5 | 34.61 | 14.41 | 37.61 | 44.88 | 44.19 | 35.58 |
| | 6 | 34.80 | 15.16 | 37.78 | 44.89 | 44.15 | 35.65 |
| | 7 | 34.93 | 15.48 | 37.85 | 44.81 | 44.22 | 35.79 |
| | 8 | 34.98 | 15.61 | 37.92 | 44.88 | 44.22 | 35.83 |
| | 9 | 35.05 | 15.78 | 38.03 | **44.77** | 44.26 | 35.87 |
| Maximum gain | | 1.22 | −5.44 | 0.77 | 8.08 | 1.94 | 5.97 |

Table 1: FID values with respect to Imagenet-100, using 50k images of Imagenet-1k, for each of the datasets and the different mixing strategies. As can be seen, MixUp improves FID by a big margin for S-21k (around 6 points), and outperforms the FID for other datasets. This is not the case for StyleGAN Oriented, which is the best previous state-of-the-art method: MixUp worsens FID, while the improvement with CutMix is marginal.

As a third interpolation strategy, we explored producing samples using the latent space of a trained generative adversarial network, StyleGAN v2 [2]. Training with samples from a GAN has been shown to improve contrastive training performance when the original data diversity is low [3]. After training StyleGAN v2 with Shaders-21k, we produce interpolated images by sampling from the network using truncation. The FID for this sampling strategy is 38.38, which improves that of the raw shaders (41.00) but falls short of simple mixing strategies (35.04 for S-21k with 6-MixUp), as seen in Table 1. We report results for MoCo training with this strategy in Section S-3.

## S-2    Finteuning experiment results

Figure 1 shows the results for finetuning a ResNet-50 trained using MoCo v2 with the datasets as described in Section 4.2 of the main paper. We finetune on Imagenet-1k for 100 epochs with a batch size of 256, starting with a learning rate of $1e-3$ and decreasing it by a factor of 10 at epochs 60 and 80. Using the different datasets as described in Section 4.2 of the main paper, we see that methods rank similarly, with S-21k performing the best overall. Although all methods perform substantially better than random initialization, differences between methods are numerically small, which motivates our choice of using linear evaluation to compare different methods instead of finetuning in the main paper.

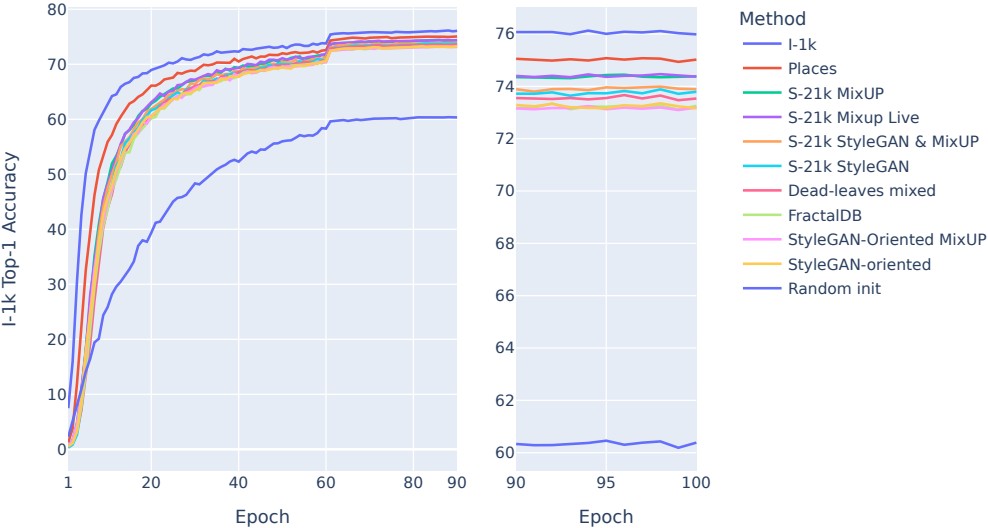

Figure 1: Performance on Imagenet-1k per epoch when finetuning a ResNet-50 pretrained on each of the datasets described in Section 4.2 of the main paper. Methods from top to bottom are sorted by final finetuning performance.

## S-3    Extended experiment results

In Table 2 we extend Table 4 of the main paper with additional mixing strategies that underperform the ones in the main paper, to validate the expected results according to the FID metric in Table 1. These StyleGAN Oriented with 6-MixUp (the previous state of the art with MixUp) and sampling from a StyleGANv2 with and without MixUp, trained on the Shaders-1k/21k dataset respectively (S-1/21k StyleGAN).

### S-3.1    VTAB detailed results

In Table 3, 4 and 5 we show detailed results per dataset on the VTAB benchmark. These show that, although on average performance correlates with Imagenet-1k/100 results, certain evaluation datasets perform differently than the average trend. This can be explained by the pre-training dataset being better aligned with the downstream task. For example, S-1/21k and the Dead Leaves images perform significantly better on dSprites than alternatives (as seen in Table 5), as these tasks consist of classifying the position and orientation of simple geometric shapes.

| Pre-train Dataset | I-1k | I-100 | VTAB Nat. | VTAB Spec. | VTAB Struct. |
|---|---|---|---|---|---|
| Random init | 4.36 | 10.84 | 10.98 | 54.30 | 22.64 |
| Places365 [4] | 55.59 | 76.00 | 59.72 | 84.19 | 33.58 |
| ImageNet-1k | 67.50 | 86.12 | 65.90 | 85.02 | 35.59 |
| StyleGAN O. [1] | 38.12 | 58.70 | 54.19 | 81.70 | **35.03** |
| StyleGAN O. [1] (MixUp) | 31.73 | 53.44 | 51.26 | 81.39 | 33.21 |
| FractalDB-1k [5] | 23.86 | 44.06 | 38.80 | 76.93 | 31.01 |
| Dead-leaves Mixed [1] | 20.00 | 38.34 | 35.87 | 74.22 | 30.81 |
| S-1k | 16.67 | 34.56 | 32.39 | 75.28 | 28.23 |
| S-1k StyleGAN | 30.68 | 51.30 | 49.70 | 79.91 | 33.06 |
| S-1k MixUp | 38.42 | 60.04 | 53.24 | 82.08 | 30.32 |
| S-21k | 30.25 | 51.52 | 45.23 | 80.75 | 32.85 |
| S-21k StyleGAN | 35.19 | 57.04 | 54.72 | 81.17 | 34.74 |
| S-21k SGAN + MUp | 36.46 | 58.40 | 53.52 | 81.47 | 32.14 |
| S-21k MixUp | **44.83** | **66.36** | **57.18** | **84.08** | 31.84 |
| S-21k MixUp Live G. | **45.25** | **66.42** | **58.20** | **84.41** | 32.25 |

Table 2: Top-1 accuracy with linear evaluation on ImageNet-1k/100 and the VTAB suite of datasets (averaged over the Natural, Specialized, and Structured categories), for a ResNet-50 trained with MoCo V2 with 1.3M samples for each of the pre-training datasets. Underlined results correspond to an upper bound (training with natural images different than the evaluation distribution) and the previous state-of-the-art without real data StyleGAN Oriented [1]. The last row (S-21k MixUp Live G.) corresponds to sampling from the shaders for each new batch, as described in Section 4 of the main paper.

| Pre-train Dataset | CIFAR | Flowers | Pets | SVHN | Caltech | DTD | Sun397 |
|---|---|---|---|---|---|---|---|
| Random Init | 10.28 | 9.30 | 7.71 | 22.02 | 14.99 | 8.94 | 3.64 |
| Places | 34.66 | 77.75 | 57.54 | 58.94 | 78.12 | 61.91 | 49.12 |
| ImageNet-1k | 48.80 | 83.51 | 71.60 | 61.30 | 81.11 | 68.03 | 46.94 |
| Stylegan Oriented | **40.91** | 70.29 | 47.42 | 62.27 | 70.30 | 55.80 | 32.36 |
| StyleGAN O. (MixUp) | 40.38 | 64.71 | 45.43 | 61.90 | 65.43 | 51.86 | 29.09 |
| FractalDB-1k | 27.52 | 51.63 | 35.73 | 39.96 | 54.27 | 42.34 | 20.16 |
| Dead-leaves M. | 24.71 | 39.60 | 23.44 | 52.67 | 52.78 | 39.20 | 18.71 |
| S-1k | 23.66 | 35.34 | 27.01 | 42.33 | 42.64 | 43.40 | 12.36 |
| S-1k Stylegan | 39.62 | 59.26 | 40.56 | 67.75 | 63.53 | 52.45 | 24.76 |
| S-1k MixUp | 32.57 | 74.19 | 50.72 | 56.88 | 67.31 | 59.68 | 31.30 |
| S-21k | 29.38 | 63.98 | 37.45 | 42.38 | 59.55 | 56.60 | 27.26 |
| S-21k StyleGAN | 39.98 | 64.95 | 45.43 | **75.41** | 71.06 | 56.81 | 29.41 |
| S-21k SGAN + MixUp | 32.13 | 69.23 | 47.18 | 68.54 | 69.67 | 58.09 | 29.81 |
| S-21k MixUp | 32.77 | **79.41** | **56.36** | 57.24 | **72.22** | **65.80** | **36.47** |
| S-21k MixUp Live G. | 34.76 | 78.99 | 56.61 | 62.62 | 73.49 | 65.21 | 35.74 |

Table 3: Top-1 accuracy for a MoCo V2 with a Resnet-50 for each of the Natural datasets in the VTAB suite, trained with a maximum of 10k samples (when more than that is available).

| Pre-train Dataset | EuroSAT | Resisc45 | Retino. | Camelyon |
|---|---|---|---|---|
| Random Init | 49.17 | 19.89 | 73.19 | 74.96 |
| Places | 91.89 | 85.96 | 74.59 | 84.30 |
| ImageNet-1k | 95.20 | 86.18 | 75.74 | 82.95 |
| Stylegan Oriented | **92.96** | 78.23 | 73.73 | 81.86 |
| StyleGAN O. (MixUp) | 92.94 | 77.02 | 73.81 | 81.80 |
| FractalDB-1k | 83.56 | 70.03 | 73.88 | 80.25 |
| Dead-leaves M. | 85.98 | 58.78 | 73.38 | 78.75 |
| S-1k | 86.81 | 60.17 | 73.22 | 80.91 |
| S-1k Stylegan | 90.67 | 74.46 | 73.34 | 81.17 |
| S-1k MixUp | 90.93 | 80.36 | 75.11 | 81.92 |
| S-21k | 90.56 | 77.75 | 74.07 | 80.61 |
| S-21k StyleGAN | 90.96 | 78.01 | 73.68 | 82.01 |
| S-21k SGAN + MixUp | 91.52 | 77.75 | 73.81 | 82.81 |
| S-21k MixUp | 92.72 | **83.68** | **75.21** | **84.72** |
| S-21k MixUp Live G. | 93.07 | 84.95 | 75.19 | 84.41 |

Table 4: Top-1 accuracy for a MoCo V2 with a Resnet-50 for each of the Specialized datasets in the VTAB suite, trained with a maximum of 10k samples (when more than that available).

| Pre-train Dataset | ClvrD | ClvrC | dSprO | dSprL | sNorbE | sNorbA | DMLab | KittiD |
|---|---|---|---|---|---|---|---|---|
| Random Init | 43.08 | 23.98 | 7.88 | 6.58 | 17.28 | 9.39 | 27.89 | 45.01 |
| Places | 48.56 | 44.57 | 13.92 | 11.71 | 35.65 | 24.47 | 44.65 | 45.15 |
| ImageNet-1k | 51.09 | 47.40 | 12.92 | 13.52 | 38.83 | 27.69 | 43.64 | 49.65 |
| Stylegan Oriented | **55.10** | **47.69** | 12.71 | 14.41 | **38.22** | 23.48 | 40.13 | **48.52** |
| StyleGAN O. MU | 54.29 | 44.90 | 12.85 | 15.78 | 34.45 | 20.56 | 37.59 | 45.29 |
| FractalDB-1k | 49.64 | 39.50 | 16.51 | 16.11 | 30.91 | 18.10 | 34.10 | 43.18 |
| Dead-leaves M. | 48.24 | 39.94 | 12.51 | **20.64** | 30.89 | 16.46 | 34.22 | 43.60 |
| S-1k | 38.95 | 31.36 | 19.01 | 16.27 | 28.17 | 15.51 | 31.96 | 44.59 |
| S-1k Stylegan | 51.68 | 39.75 | 15.92 | 17.98 | 33.60 | 21.52 | 38.33 | 45.71 |
| S-1k MixUp | 47.33 | 42.70 | 11.17 | 10.81 | 29.29 | 17.77 | 36.90 | 46.55 |
| S-21k | 47.48 | 41.13 | 18.33 | 18.74 | 35.13 | 19.22 | 37.64 | 45.15 |
| S-21k StyleGAN | 51.28 | 42.64 | **19.05** | 17.86 | 37.09 | **26.70** | 40.39 | 42.90 |
| S-21k SGAN + MU | 50.85 | 42.41 | 13.55 | 16.71 | 29.66 | 20.55 | 39.79 | 43.60 |
| S-21k MixUp | 48.07 | 45.42 | 11.69 | 13.96 | 29.92 | 19.90 | **41.56** | 44.16 |
| S-21k MixUp Live G. | 47.29 | 43.62 | 12.17 | 15.78 | 34.29 | 20.78 | 40.07 | 44.02 |

Table 5: Top-1 accuracy for a MoCo V2 with a Resnet-50 for each of the Structured datasets in the VTAB suite, trained with a maximum of 10k samples (when more than that is available).

## S-4  Nearest Neighbor retrieval

In Figures 2 and 3we show additional 5 nearest neighbors retrieval results (sampled at random) for our network and several baselines, that complement the results in Figure 4 of the main paper. These show that our best-performing method retrieves qualitatively better results than previous methods and simple baselines, and the performance gap compared to training with real images is greatly reduced.

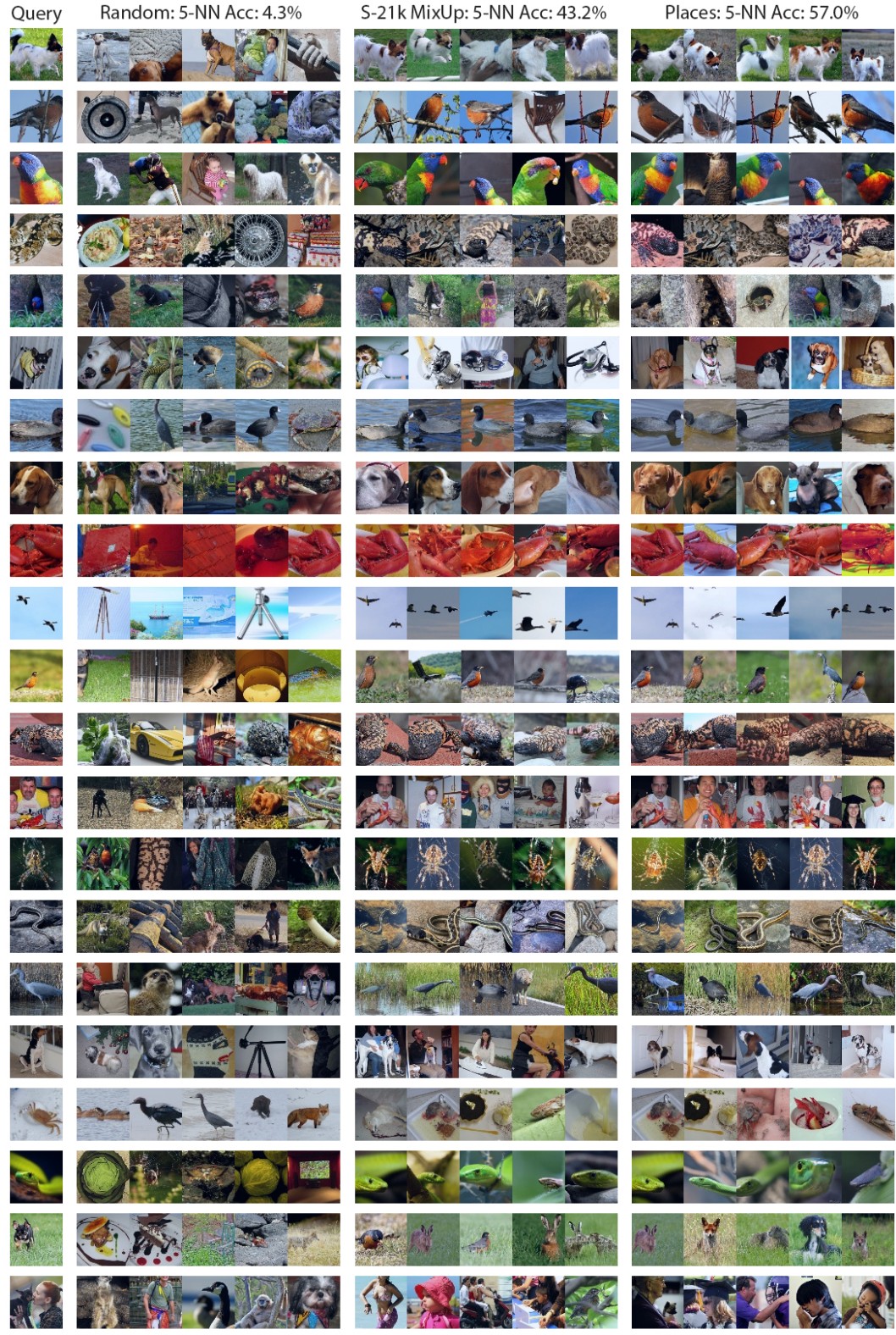

Figure 2: 5 nearest neighbors on ImageNet-100 for ResNet-50 randomly initialized (left) or trained with MoCo v2 on Shaders-21k with MixUp (middle) and Places (right). Reported accuracy corresponds to 5-NN accuracy on ImageNet-100 and queries have been selected at random.

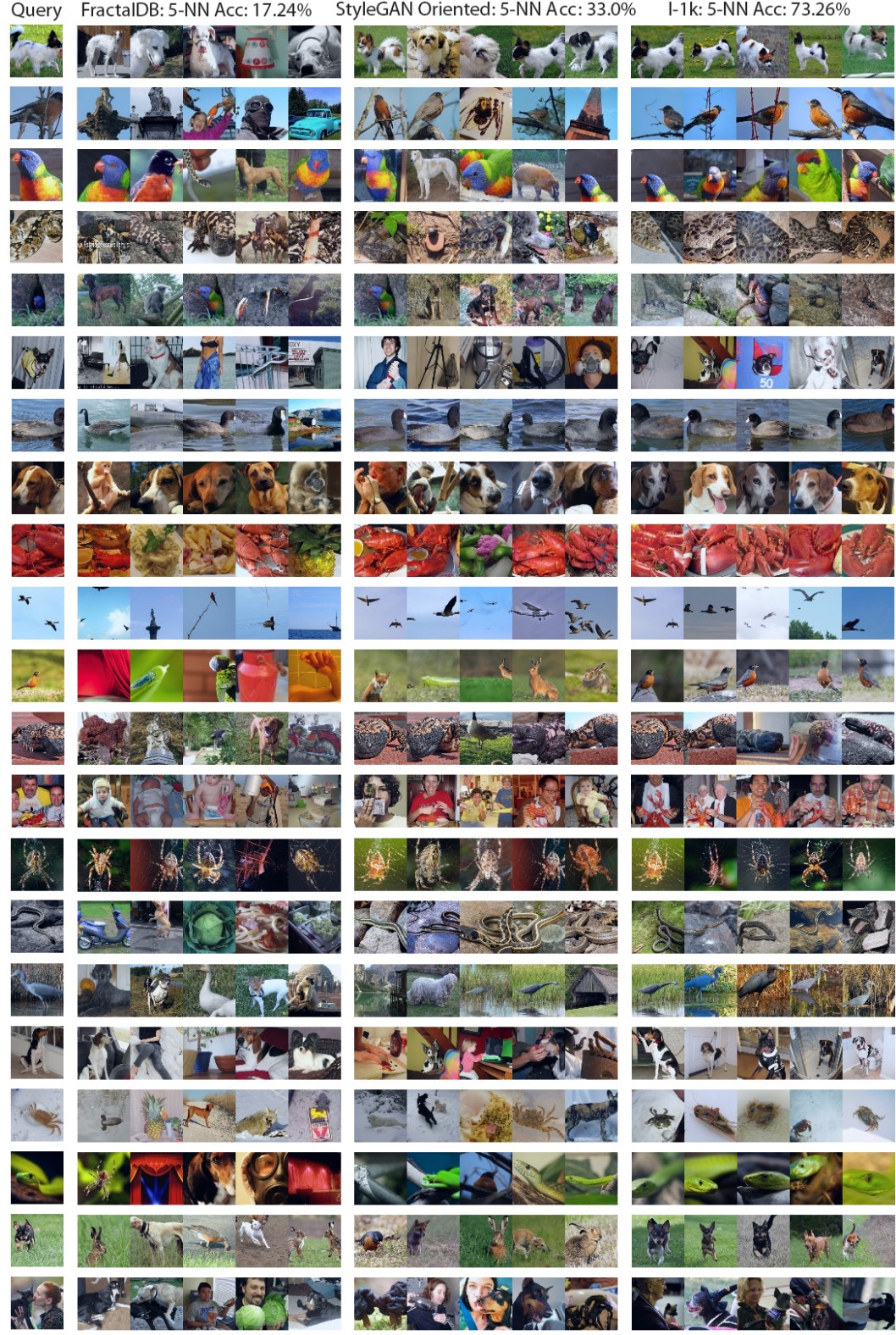

Figure 3: 5 nearest neighbors on ImageNet-100 for a ResNet-50 trained with MoCo v2 on FractalDB (left), StyleGAN oriented (middle) and Imagenet-1k (right). Reported accuracy corresponds to 5-NN accuracy on ImageNet-100.

## S-5 Feature visualizations

Figures 4-8 show feature visualizations for different units of several layers of a ResNet-50 using the method in [6].

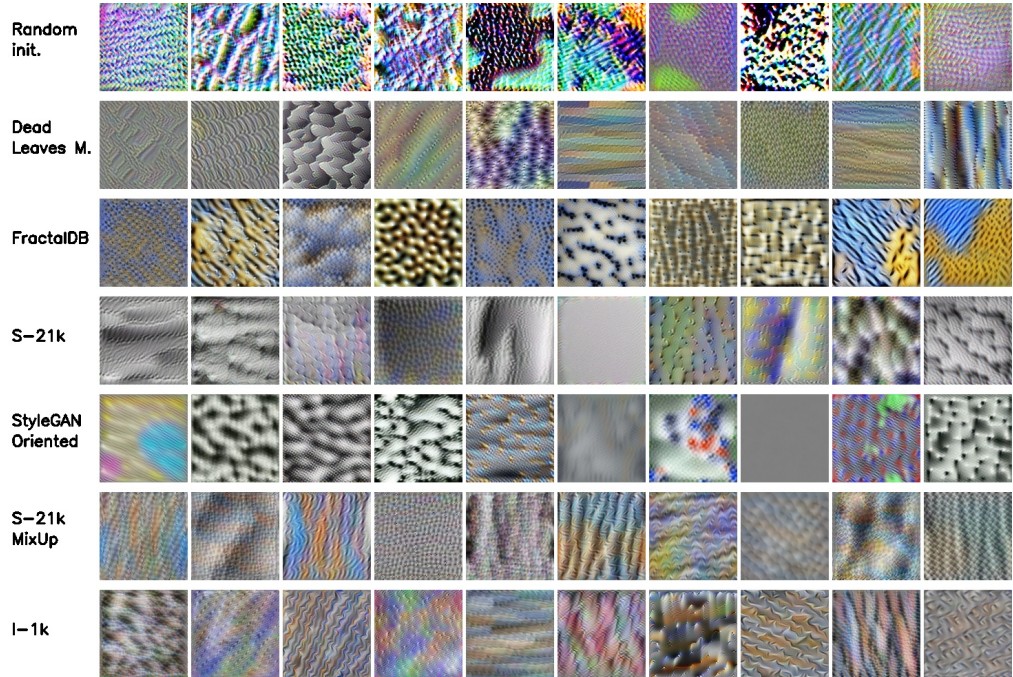

Figure 4: Feature visualizations for random units at layer1_2_conv3 of a ResNet-50 trained with several of the datasets described in Section 4 of the main paper, using the method in [6]

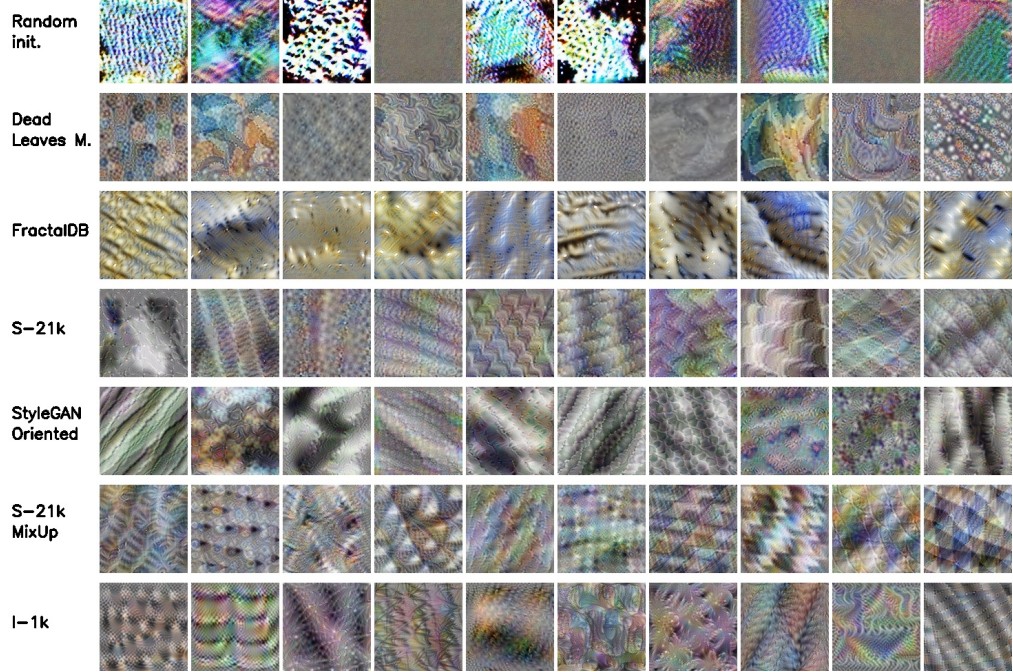

Figure 5: Feature visualizations for random units at layer2_3_conv3 of a ResNet-50 trained with several of the datasets described in Section 4 of the main paper, using the method in [6]

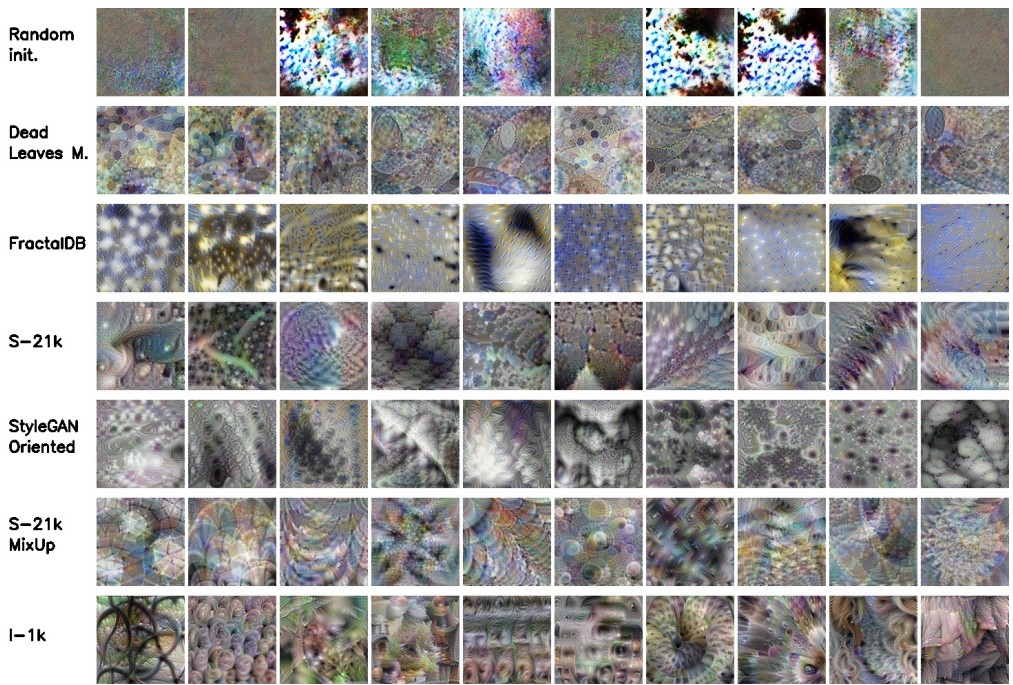

Figure 6: Feature visualizations for random units at layer3_5_conv2 of a ResNet-50 trained with several of the datasets described in Section 4 of the main paper, using the method in [6]

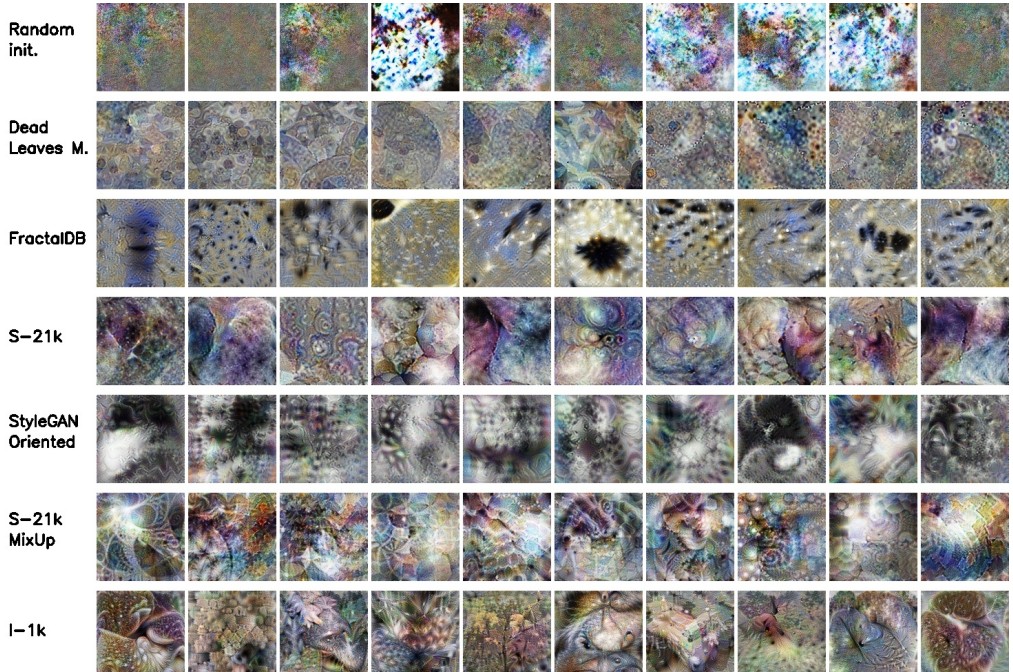

Figure 7: Feature visualizations for random units at layer4_2_conv3 of a ResNet-50 trained with several of the datasets described in Section 4 of the main paper, using the method in [6]

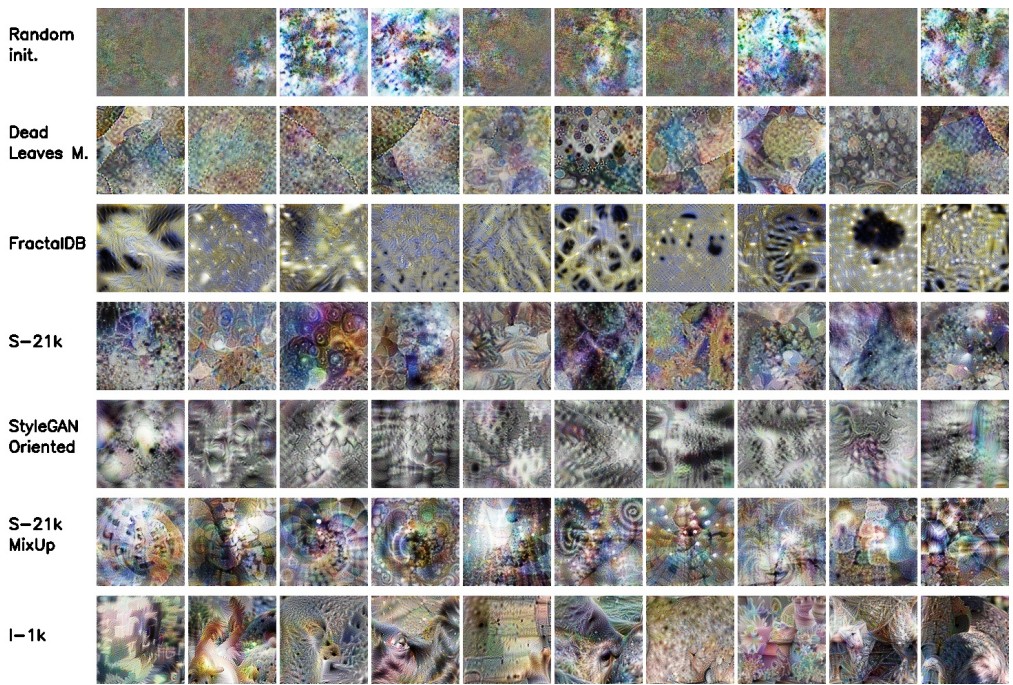

Figure 8: Feature visualizations for random units at the fully connected projection layer of a ResNet-50 trained with several of the datasets described in Section 4 of the main paper, using the method in [6]

## S-6 Dataset Samples

### S-6.1 S-1k

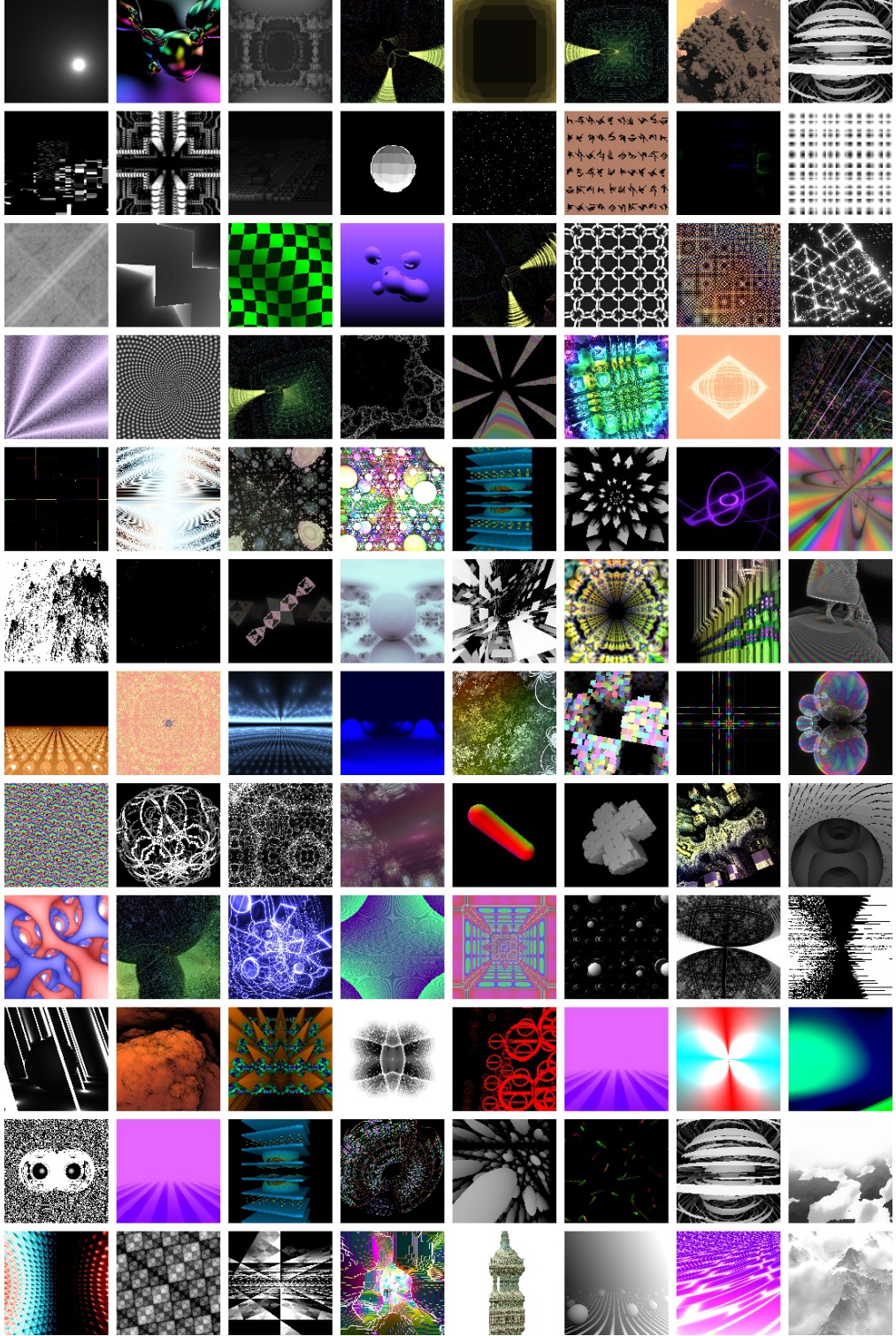

Figure 9: 96 random samples of the dataset S-1k.

## S-6.2    S-1k StyleGAN

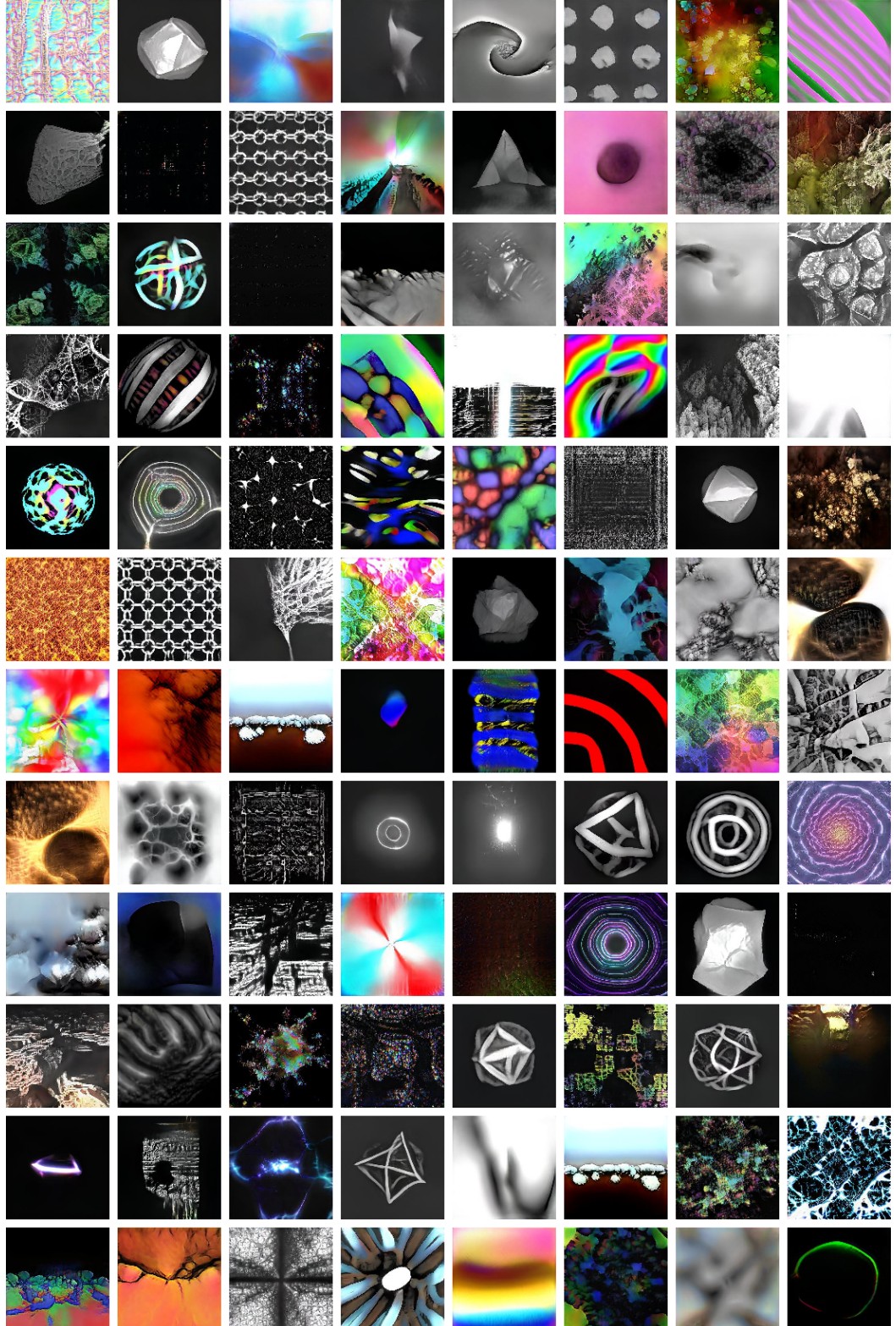

Figure 10: 96 random samples of the dataset S-1k StyleGAN.

## S-6.3 S-1k MixUp

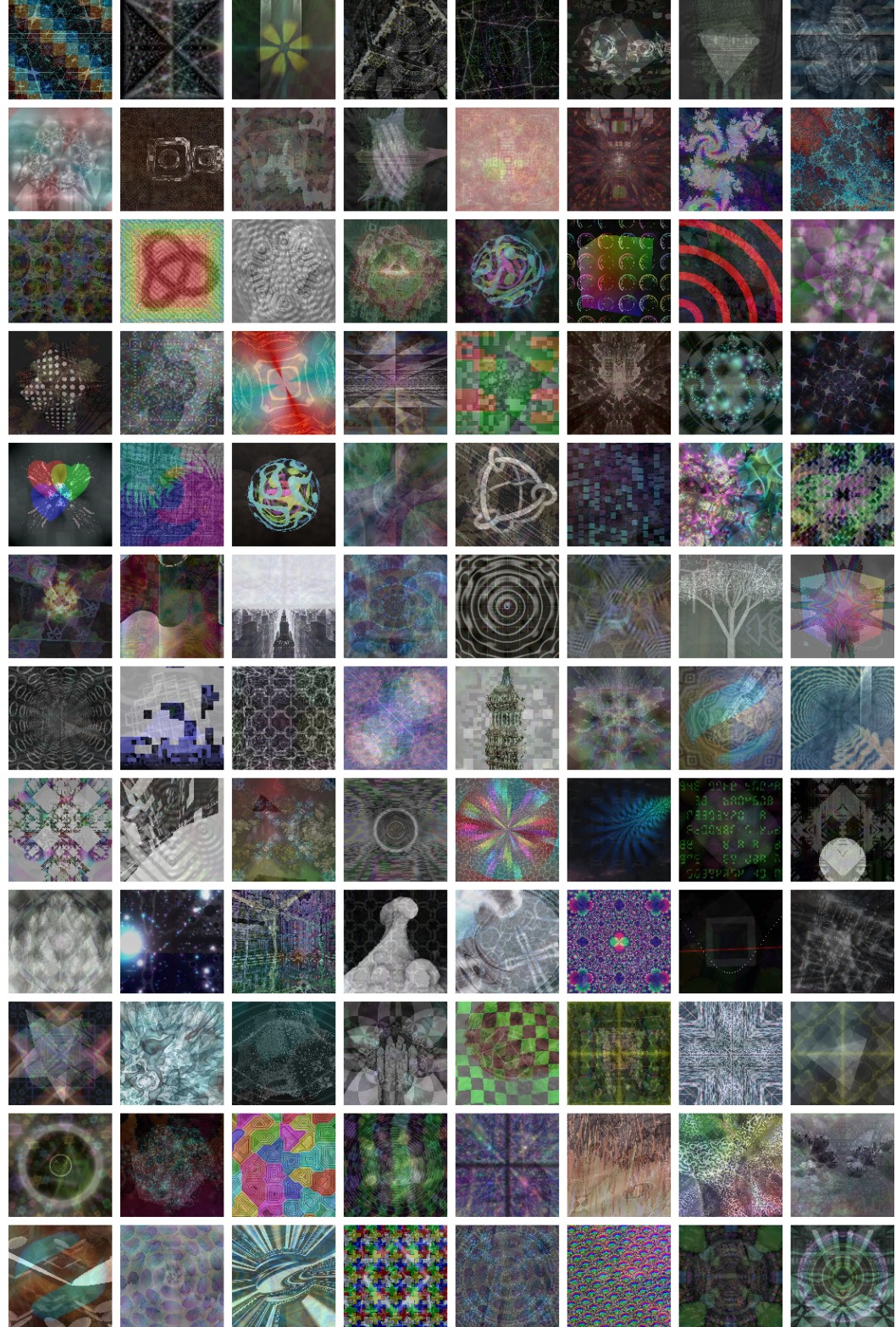

Figure 11: 96 random samples of the dataset S-1k MixUp.

**S-6.4    S-21k**

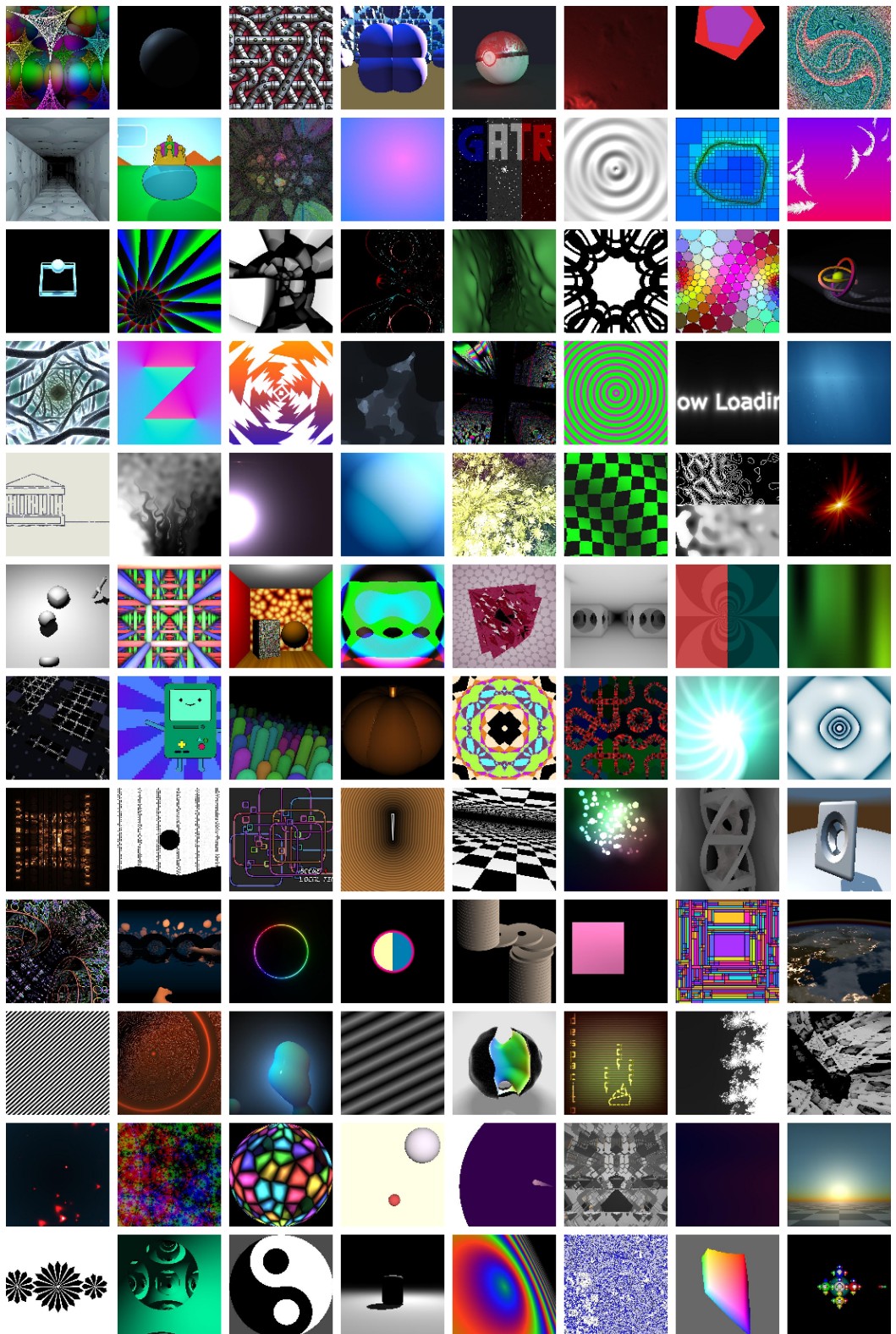

Figure 12: 96 random samples of the dataset S-21k.

### S-6.5 S-21k StyleGAN

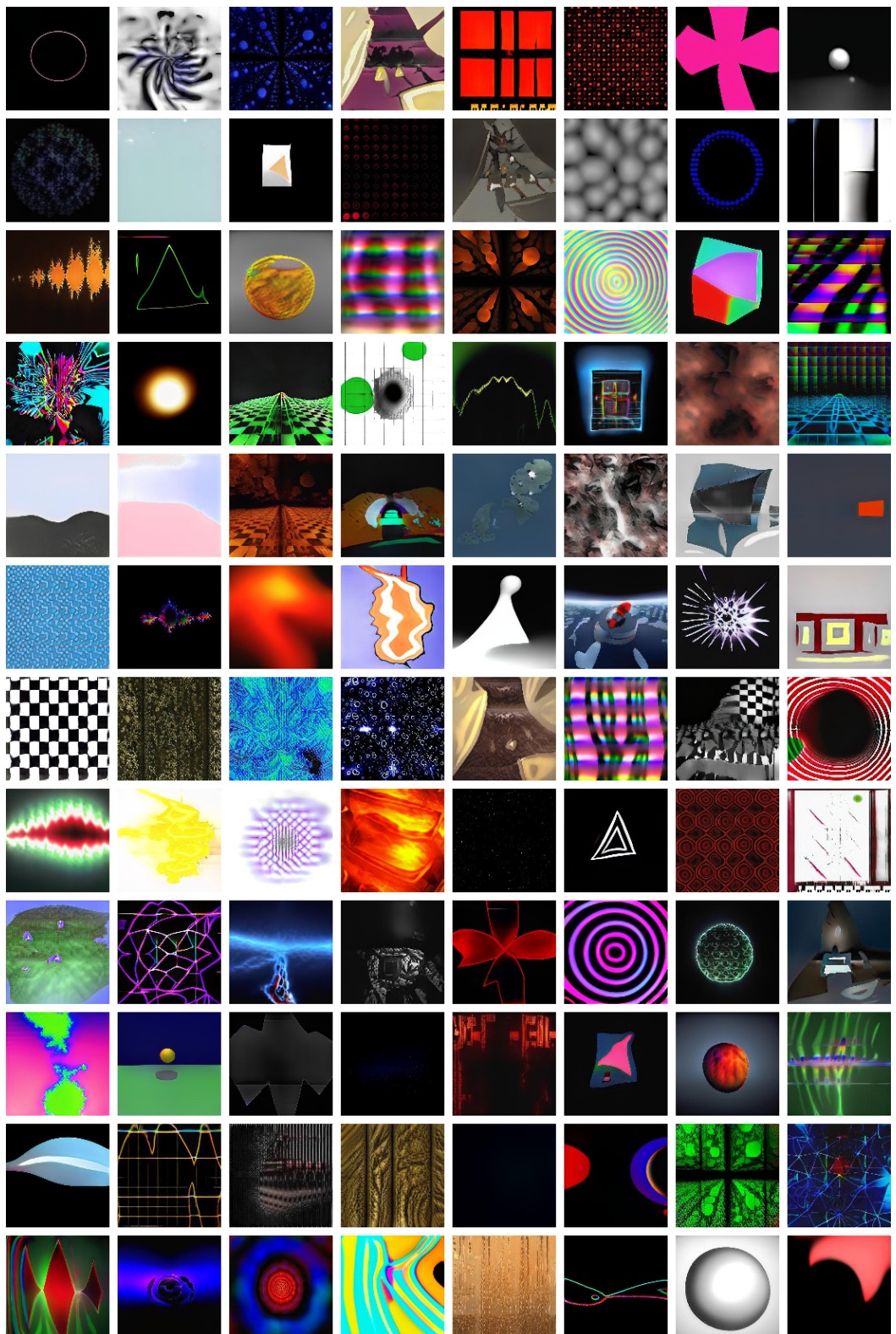

Figure 13: 96 random samples of the dataset S-21k StyleGAN.

**S-6.6    S-21k MixUp**

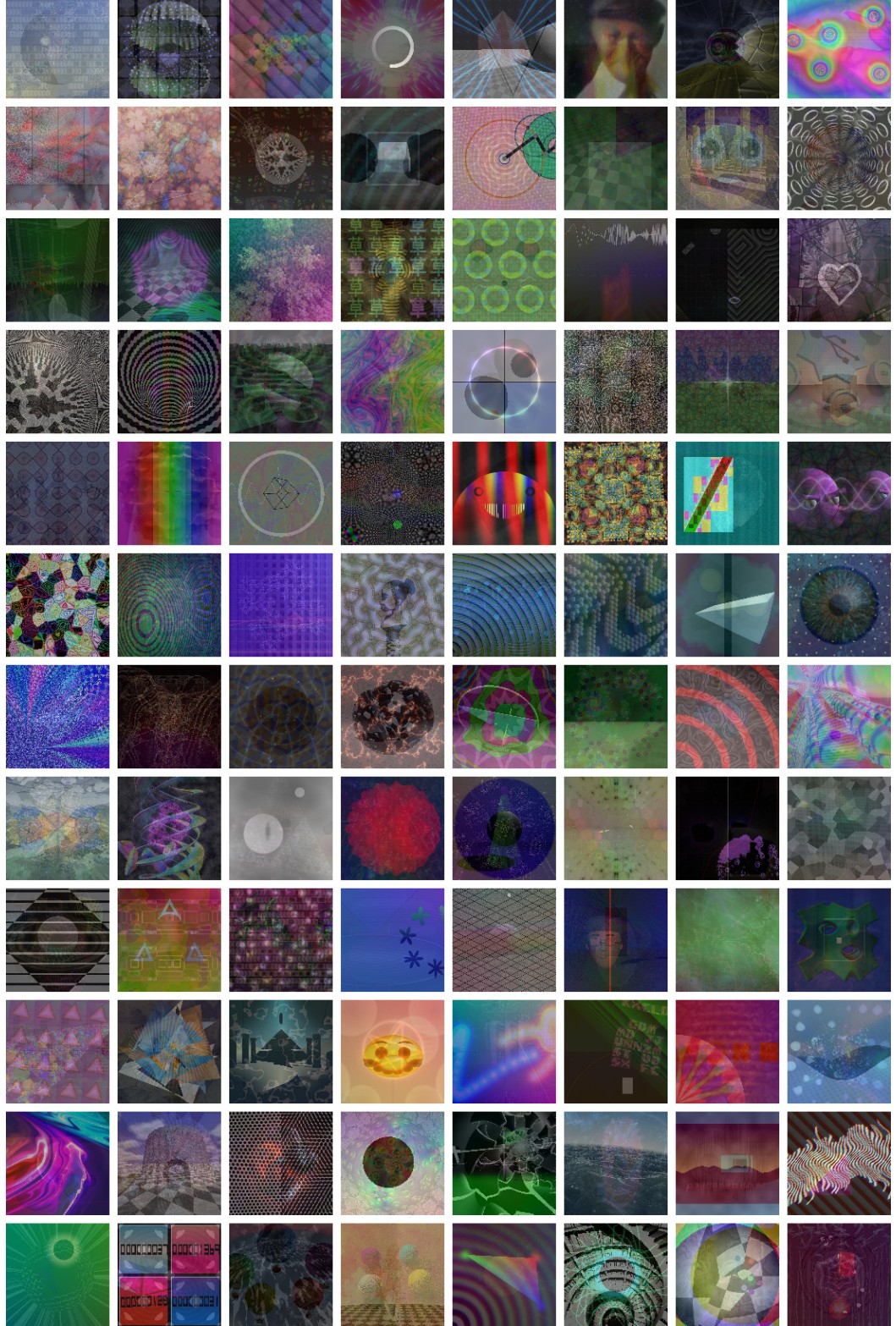

Figure 14: 96 random samples of the dataset S-21k MixUp.