# OpenReview forum: "Procedural Image Programs for Representation Learning"
_NeurIPS.cc/2022/Conference — NeurIPS 2022 Accept_

### Official Review · Reviewer_CTGk · 2022-07-09

**Rating:** 6
**Confidence:** 4
**Soundness:** 3 good
**Presentation:** 3 good
**Contribution:** 3 good

**Summary:**

This paper proposed a dataset for learning representations without access to real images, the dataset is composed of twenty-one thousand OpenGL programs is proposed for supervised and unsupervised representation learning, the synthetic data can help reduce the concerns about using real images in training such as human face etc.
Experiments show that pretraining on the proposed dataset can bring performance improvement over previous synthetic dataset, and the investigation on what makes for a good generative image program seems interesting.


**Questions:**

Q1: It would be good if the paper could also include some visualization results of the learned representations like in [R1], and maybe the linear readout performance of the representations per layer? I think this could give the reader more sense of how well the learned representation is.

Q2: Should the augmentations on synethic data be the same as real data? IMO, the data augmentations used by MoCov2 on ImageNet is choosen because the augmentation can encode some prior knowledge about the real images, so probably when we are training the model on synthetic images, the choice of augmentations should be different.

Q3: There are efforts of reducing privacy concerns in the real data such as PASS [R2], it would be good if the paper could discuss the connection to these types of works and what is the different between using synthetic images and modified real images.

Q4: It would also be interesting to see how well the learned representation on the proposed dataset transfers by finetuning the representation on a target dataset, because as shown in [R3], the linear readout performance does not always correlates with the full finetuning results.

[R1] A critical analysis of self-supervision, or what we can learn from a single image, ICLR 2020

[R2] PASS: An ImageNet replacement for self-supervised pretraining without humans, NeurIPS 2021 Datasets and Benchmarks

[R3] Masked Autoencoders Are Scalable Vision Learners, CVPR 2022


**Limitations:**

1. There is no section about the limitation and negative societal impact in the main paper and the supplementary, but IMO learning representation on synthetic data does not bring any negative impact in general, however, regarding this work, I am curious about the license of the all the shader used in the dataset, do all of them allow for research purposes?
2. One limitation for learning representation using synthetic data is the performance is not comparable with training on real images, but I think by studying what factors help learning a good representation using synthetic data, this paper is a meanful step towards makeing representation learned on synthetic data usable.


**Strengths And Weaknesses:**

S1: The goal of learning representation without real images is practically important.

S2: The analysis of the properties for good image generation programs could be of interest for people who are working on image generation programs for representation learning.

W1: The evaluation of the learned representations on the proposed dataset is only limited to linear readout, while I believe using full finetuning can better show the difference between representations and are closer to a real use case.

W2: There are works that aims to learn representations using only one image [R1] or using filtered real images [R2], I think the paper should include discussion with these works because the goal is the same: learning the representation without ethical concerns.

---

> ### Author Response · Authors · 2022-07-28
> **For reviewer CTGk**
>
> **W1)** Although fine tuning has important practical applications (specially to achieve SoTA performance, which this work does not focus on), as pointed out in [RE1, RE2] a carefully designed training schedule from a random initialization can perform as well as finetuning. This is true even for tasks were the previous general consensus was that pre-training greatly improves performance: in the case studied in [RE1], pretraining for classification on Imagenet and then finetuning for object detection. Consequently, certain hyperparameter settings could make the finetuning performance for our baseline (random initialized network) and our upperbound (real images) to be close or even the same. In a similar manner, particular settings of the finetuning hyperparameters could make our models perform as well as the upperbound (or even be numerically better because of randomness during training), despite them not having learnt good image representations.
>
>
> On the other hand, linear probing and k-NN are simpler testing schemes that keep most (or all) the parameters fixed. They perform substantially differently with a random network than with one trained on Places (4.36% vs 55.59% top-1 acc in the case of linear probing on I-1K). Consequently, this testing scheme is less sensitive to hyperparameter tuning and allows to clearly measure progress when evaluating different image models, making sure differences are not caused by a given hyperparameter setting.
>
> **W2)** We performed initial experiments using the three images on [R1] Figure 1 and standard contrastive techniques at small scale (using SimCLR with an Alexnet and 64x64 resolution images). For images A, B and C in Figure 1 of [R1], SimCLR pre-training with Imagenet-100 performed worse than a random initialized baseline (0.13, 0.18 and 0.12, respectively, compared to 0.20 for a randomly initialized one). We do not include these negative results in the paper, as other single images or training algorithms may perform better.
>
> We will include a paragraph on the related work section to discuss [R1], [R2] and similar efforts. Although works such as [R1] and [R2] greatly reduce the privacy concerns of image datasets (in the case of R2 for human subjects present on Imagenet), our method and similar approaches go one step beyond: not use any real images at all.
>
> **Q1)** Through our experimentation, we have found feature visualizations to give lots of insights for the learnt representations, and we will include them in the supplementary material. Linear read out performance per layer decreases for shallower layers, and at the same time becomes closer to the performance when training with real images at the same layer. As pointed out by works like [2,13] representations learnt at shallow layers are more similar (and as good) throughout datasets.
>
> **Q2)** Indeed, MoCo v2 and the augmentation as they are may not be the optimal training paradigm for this source of data. For example, [RE3] points out that it is crucial to recompute batch normalization statistics when pretraining uses noise-like samples to close the domain gap, and we do not do this. We take the decision of keeping the algorithm fixed to make sure that we are measuring progress towards closing the gap between synthetic and real images. Changing the training setup for synthetic data while keeping the training setup for real data fixed is not a fair comparison.
>
> **Q3)** See W2)
>
> **Q4)**  See W1)
>
> **L1)** The data has been obtained through two data sources: Shadertoy (20k/21k programs) and Twigl (1k/21k). The general license of a shader in the webpage is CreativeCommons BY-NC-SA 3.0, which allows for research. For twigl, the general Twitter copyright applies. Still, to preserve content attribution we will release the dataset as urls to the original sources.
>
> **L2)** Although the performance is worse than training with real images, we 1) show that it outperforms baselines and previous methods 2) we provide a scaling study on how performance improves with the number of programs, which points out a clear direction on how to further close the gap: obtain more samples of our distribution (i.e. more uncurated programs). This is in contrast to previous works, where obtaining more samples from their proposed distribution (i.e. more images of their curated  programs) saturates performance fast, thus providing no clear path to improve performance.
>
> [RE1] Rethinking ImageNet Pre-training, Kaiming He, Ross Girshick, Piotr Dollár
>
> [RE2] Are Large-scale Datasets Necessary for Self-Supervised Pre-training? Alaaeldin El-Nouby, Gautier Izacard, Hugo Touvron, Ivan Laptev, Hervé Jegou, Edouard Grave
>
> [RE3] Pure Noise to the Rescue of Insufficient Data: Improving Imbalanced Classification by Training on Random Noise Images. Shiran Zada, Itay Benou, Michal Irani

---

> > ### Comment · Reviewer_CTGk · 2022-08-03
> > **Thanks for the rebuttal**
> >
> > Thanks to the author for the rebuttal.
> > After reading the rebuttal and the reviews from the other reviewers, I have the following questions and suggestions for the author:
> >
> > Regarding my question Q4, I agree with the rebuttal that carefully designing the training schedule could improve the performance even for random initialized models, but this does not answer my question: **Do the performance of the representations learned on the proposed dataset still close to the representations learned on real images after fine-tuning?** If the hyper-parameters are the same, the comparison would still be fair, and would give more insights into the properties of the representations on these synthetic images. Also as the rebuttal says, fine-tuning has more important practical applications, so I think this is one of the important thing to understand about this work. Furthermore, although fine-tuning with carefully designed training schedules could improve the performance, the point of pre-training is to get rid of this trial-and-error process, and can achieve good performance without fine-tuning the hyperparameters (Otherwise just using a random initialization for every application and then fine-tune the training will have no ethical issues for pre-training at all).
> >
> > Regarding my question Q2, I agree that fixing the training setup for synthetic and real data is a fair comparison, but this does not contribute to the overall goal: Closing the gap between representations learned on synthetic data and real data. The main question is, can you provide some hint about how to modifiy the training setup for the synthetic data so that the final performance would be close to the representations on the real data? This question will not be considered for my final recommendation, because this could be another paper, but I think at least acknowledge or give a simple attempt on this direction in the paper would make the submission much stronger.
> >
> > From the weaknesses listed by reviewer vdrw and the rebuttal submitted by the author, I have one question: the rebuttal says that generating more and diverse samples that have similar statistical properties to natural images helps pre-training, so what about using images renderred by some game engines such as blender rather than using shaders? Could blender renders more real-like images and helps the performance furthermore? This question also will not be considered in my final recommendation, but I would like to see what's the opinion of the authors.
> >
> > Minor:
> >
> > I hope the author would include the limitation and weakness part in the revised version of this paper.

---

> > > ### Author Response · Authors · 2022-08-03
> > > **Extra clarifications**
> > >
> > > **Q4 extra clarification**: Despite the justification on why we focus our study on linear probing that follows, we agree with the reviewer that readers may find finetuning results informative, and we will include results for finetuning (for different number of epochs) on Imagenet-1k/100 for the models in Table 3 in the revised manuscript.
> > >
> > > During experimentation, we tested finetuning in some restricted settings, and found that 1) final performance correlates with linear probing performance and 2) performance differences between methods are small and hard to interpret. For example, training MAE with a restricted computational budget (50 epochs of training during both train and evaluation) performs as follows for I-100. For a model randomly intialized, trained with S-1k w/o mixup, StyleGAN-Oriented (previous SoTA) and S-1k w/ mixup performance with linear probing is **12.5**, **25.1**, **38.0**, and **46.2** respectively. When finetuned, the performance is **68.6**, **89.0**, **87.2** and **90.7**. As can be seen, the difference between the first and second best performing method is only **1.7%** when finetuning, while it is **8.2%** when linear probing. It is also expected that this difference will become even smaller with extra finetuning. This can be seen as the random baseline is far from convergence (the 68.6 performance is still low for training from a random initialization on I-100) while linear probing is is already close to convergence (comparable results to the ones seen in Figure 3). To make finetuning results interpretable, they are typically reported at different epochs [1,13] (which require different runs if using warmup or cosine annealing for the lr)  and/or with different parts of the network being optimized [24]. This would multiply by N the total number of experiments to run, and at the same time lead to less conclusive results.
> > >
> > > Because of computational constraints, performing all experiments described in the paper (20 for Table 2,  17 * 3 = 51 for Figure 3, 11 * 21 dataset = 231 runs for Table 3, 1000 runs for experiments in Figure 5, 33 in Figure 7, for a total of roughly 1300 runs) with different eval setups is unfeasible. We perform them on linear evaluation, because of the aforementioned reason.
> > >
> > > **Q2 extra clarification)** In the original rebuttal, we mention [RE3] as a way to improve performance. Well-known domain adaptation techniques will also improve performance, as the training and eval/test domains are different. Another simple way to improve performance is searching for the best hyperparameters of MoCo with grid search (adjusting the levels of crop/color jittering/..., training schedule, optimizer...). The ones we use have been carefully designed for Imagenet-1k, and we don't modify them to 1) avoid settings were we may be favoring our processes compared to real images (which would be unfair) 2)  keep consistent hyperparmaters to make results comparable to real images. Furthermore, our scaling studies point out that more samples (in our case, programs) are likely to improve performance, while more samples from previous methods (images) showed that performance saturates fast [2]. These are clear avenues that one could pursue to close the gap between real and synthetic images. As these should hold in general for any of the previously proposed synthetic processes, studying these avenues is a different paper in itself. Despite this, we will add a discussion of this in the limitations section.
> > >
> > > **[From the weakness listed...]** Yes, blender renders could improve performance and they can be seen as a (much more complex) procedural program. They have serveral practical nuisances compared to shaders, which include:
> > > - They are less compact in disk/memory (shaders occupy 11KB on disk on average, compared to several MB of typical 3D models)
> > >
> > > - Rendering while training is computationally much more expensive (see Table 1). On our implementation, an average shader outputs 979.66 FPS, most of the time being spent on memory transfer, which could be avoided by sharing the memory between the training and rendering process (we have not implemented this, as this throughput is enough for our study). All the shaders compiled take ~40MB on GPU memory, while 3D models take much more GPU memory.
> > >
> > > - How to sample from them needs to be engineered (i.e how to take different and diverse renders given a 3D model)
> > >
> > > - They are designed to mimic reality and capture more biases than shaders.
> > >
> > > Despite this, the strategy we propose for shaders (collecting them at scale instead of curating a few good ones) could also be applied to 3D models, and readers should expect similar conclusions as the ones shown in the paper for shaders to hold.

---

> > > > ### Comment · Reviewer_CTGk · 2022-08-04
> > > > **Thanks for the extra clarification**
> > > >
> > > > **Q4 extra clarification**: Thanks for performing the experiments, and I think the results and the argument makes sense. I would also suggest including this in the revision of the paper.
> > > >
> > > > **Q2 extra clarification**: I agree that searching for the optimal hyperparameters could be helpful, future works could also design novel methods for this purpose, and I think adding this discussion to the paper can make it stronger.
> > > >
> > > > **Blender**: Thanks for the clarification, the argument makes sense to me.

---

### Official Review · Reviewer_vdrw · 2022-07-12

**Rating:** 5
**Confidence:** 3
**Soundness:** 2 fair
**Presentation:** 2 fair
**Contribution:** 2 fair

**Summary:**

Manuscript presents a method to train neural networks over synthetic data. The work collected a large collection of 21K OpenGL programs for rendering diverse set of synthetic images to be used for representation learning. Experiments show that the collected programs can be used for pretraining in both supervised and unsupervised fashion. New state of the art results are obtained for pretraining with procedurally generated images and are observed to compete closely against pretraining with real data samples.

**Questions:**

- While the authors clearly mention that their approach involves generating images from several processes (but with fixed $\theta$), it is not clearly described how it is different from the existing works [1, 2] which focus on a single program ($i=1$) but with varied parameters $(\theta)$ ? In other words, although it may be intuitive to see that more processes are better than a single process, how to quantify (even loosely) the bigger or better quotient given the degrees of freedom are different ($\theta$ versus $i$ in lines 94-100).

**Limitations:**

- As discussed in the weaknesses section, the major limitation of the work is that fails to go beyond the collection of large-scale image generating programs. Authors have attempted to study the collection and draw useful properties for crafting such programs that are more useful, but in my opinion that needs more work.
- Outright I do not see any adverse impacts of the present work on the society.

**Strengths And Weaknesses:**

Strengths
- Collected a large-scale repository of image programs for easily generating variety of synthetic images. These data samples can help pretraining while avoiding the unfair biases and other contaminations that arise from natural datasets.
- Because of the large scale, the collection enables to study how the performance scales with the size and variety of the synthetic data.
- Representation learning experiments have been performed in both the supervised and unsupervised scenarios.

Weaknesses
- The major contribution of the manuscript is somehow not very strong (despite being very important). In other words, in my opinion, large-scale collection of procedural image programs alone may not be sufficient enough to excite the NeurIPS community.
- It is easy and trivial to see that more and diverse set of synthetic data samples improve the pretraining quality. They can outperform the existing small-scale collections and close the gap between the performance of synthetic and real image datasets.
- I believe studying the effectiveness of the individual programs would be an interesting and useful contribution. The presented analysis on the properties of the programs that leads to better learning is not very conclusive or effective. In the end the reader may not feel very clear about the desired properties (except for weak observations such as LPIPS, more colors, etc.) and the study feels incomplete or lacking.

---

> ### Author Response · Authors · 2022-07-28
> **For reviewer vrdw**
>
> **W1&2)** It is well established that *more and diverse* samples that have similar statistical properties to natural images (not only *more and diverse*, e.g Gaussian noise) improve pretraining. To achieve this goal with synthetic image programs, previous works 1) curate a single or a handful of programs (fractals, dead-leaves…) and 2) carefully set the generation hyperparameters to produce samples with these properties.
> In our paper, we show a different approach which is more effective: using a set of uncurated programs (a random one underperforming any of the ones found in previous works by a big margin, as seen in Figure 3) and generating more and diverse samples by scaling up the number of programs, instead of having an expert practitioner carefully modifying them. This is one of the major contributions of the paper, which we do not believe to be an *easy and trivial* insight. Lots of effort by computer vision practitioners has been put in previous works (which we outperform) to carefully set the generation hyperparamters  (which we do not) of their curated handful of programs.
>
> Furthermore, our scaling study shows that performance is not saturating with the current number of programs. This provides a clear approach to continue improving performance: collect or learn to generate more samples in the same distribution of our collection of uncurated image programs. On the other hand, previous work has shown that generating more samples that look like the ones in their distribution (novel images from their curated programs) saturates performance fast [2].
>
> These insights are also main contributions of the paper, and we believe them to be of equal (if not of more) interest to the community as the dataset (as explained in the introduction).
>
>
> **W3)** The main result of our paper is an important finding about what makes for a good generative image program: a diverse set of uncurated programs are a better generation process than a single (or handful) programs with handcrafted diversity, which is contrary to previous long-standing approaches. Furthermore, we empirically *study the effectiveness of the individual programs* in Section 5) A single shader and show qualitative and quantitative measures that correlate (or not) with performance, which give empirical insights of what properties make for a good single program. We agree with the reviewer that understanding what properties make for good generative image programs is an important research topic, and this paper as it stands will allow for future research in this important direction.
>
>
> **Q1)**  We use this terminology to illustrate that previous works have handcrafted how to design $\theta$ to achieve good samples (which they explain in detail in the respective papers), while we use an orthogonal approach. In our case, we collect different programs all together, with varying degrees of performance (as seen in Figure 5), and do not engineer the numerical constants $\theta$ of our programs. Although our collection of programs could be seen as a single one, where what program is rendered is decided by the random variable $z$, we use this terminology to illustrate the difference with respect to previous works. We will make this terminology decision clearer in the revised manuscript.

---

> > ### Comment · Reviewer_vdrw · 2022-08-07
> > **Thanks for the rebuttal**
> >
> > Thanks to the authors for providing their responses to the reviewer comments.
> >
> > W1&2) [With all due respect to the authors] I am not totally convinced that "beating the requirement of careful (inefficient?) efforts of earlier works by the scale (alone?)" is insightful enough for the NeurIPS community. I also understand that these notions are subjective.
> >
> > However, after reading other reviews and corresponding responses I am increasing my score to 5.

---

### Official Review · Reviewer_mJZE · 2022-07-27

**Rating:** 3
**Confidence:** 1
**Soundness:** 2 fair
**Presentation:** 1 poor
**Contribution:** 1 poor

**Summary:**

This paper studies procedural image programs. Procedural image programs are image representation learning methods based on synthetically generated unnatural images (e.g., fractals and noise). The authors collected OpenGL fragment shader codes to generate diverse synthetic images and trained their image representation model on this dataset. This paper evaluated the proposed method on ImageNet and verified that their approach outperforms existing other unnatural image generation methods.

**Questions:**

I left questions in the "Strengths and Weaknesses" section.

**Limitations:**

The authors have adequately addressed the limitations and potential negative social impact of their work.

**Strengths And Weaknesses:**

Strengths

1. This paper aims to train image representation models without natural images, a challenging research problem.

Weaknesses

1. The problem statement in the introduction paragraph is that using real images causes privacy and bias problems (lines 1-2). However, this paper only shows the performance of their approach on the ImageNet dataset. Please provide more experimental results that support the original claim, alleviating the privacy problem and the bias problem.
2. This paper is hard to follow. Please add more descriptions
    1. Please provide more descriptions of existing approaches, such as the Fractal and other image generation methods. As the research topic of this paper covers unfamiliar concepts and methods, more description for background knowledge is required.
    2. Section 3.1 is hard to follow. What is the time paramter, t?
3. More experimental results are required.
    1. What happens if we use only a part of the collected programs? Can we categorize the programs? Can we measure the performance in each category? Is the Fractal method included in the collected programs?

---

> ### Author Response · Authors · 2022-07-28
> **For reviewer mJZE**
>
> **W1)** In our paper, **we show performance on 21 dataset-tasks** (**not *only on the ImageNet dataset* )**: the 19 tasks contained in the VTAB benchmark (see L178-180, L219-227, L285-287 and references to VTAB throughout the paper and Supp.Mat.) and Imagenet-1k/100. Detailed performance per dataset on the VTAB benchmark (Table 3 and Tables 3-5 of Supp.Mat.) illustrates that Imagenet pretrained models perform particularly well on animal/plants datasets (Flowers/Pets) while our models perform similarly to Places/Imagenet for arbitrary domains (SVHN, DTD, EuroSAT, Retino, Camelyon). This shows that the models pretrained on Imagenet are biased to the classes found on Imagenet, while our models perform consistently across datasets.
>
> These biases can also be qualitatively seen using feature visualizations (like the ones present in previous works like [2]), where detectors for common classes, like dogs and cats, appear when training with Imagenet. When training on our datasets, detectors correspond to geometric shapes and textures, making them less biased to the particular classes found on the pretraining dataset. We will include feature visualizations in the revised manuscript.
>
> By design, our pretrained networks do not inherit the privacy and bias concerns of real image datasets (as we don’t use them for pretraining). Despite this, bias and privacy concerns may arise during the usage of these networks for downstream tasks, as real data is then required. As our models perform well with just a linear layer or NN on top of the learnt representation, dataset biases for the downstream tasks may be easier to tackle (e.g. with linearized methods) than when the full model is biased. How to accomplish this is itself a research field and is outside of the scope of this paper.
>
>
> **W2)** We include references to previous works with similar approaches, but we do not include details of their methods. To understand our paper, they can be simply viewed as 1) one or a handful of image generation programs that 2) have been carefully tuned to produce good images for training. Our approach contrasts with previous works by using uncurated programs at scale, which are not tuned at all. As one of the points of our paper is not to manually tune these programs and instead focus on scaling the number of programs, the details of the previous generative programs are not relevant for our work.
>
> The variable t is a stochastic variable that is present in the collection of shaders, and can be sampled to produce (limited) diversity per shader. This is described in detail in lines L115-119.
>
>
> **W3)** Scaling studies (i.e. *What happens if we use only a part of the collected programs?*) is one of the main contributions of the paper (first point of the main contributions in Sec 1). This is studied in detail in Sec 4.1 (see Fig 3) and Sec 5) A collection of shaders, where we study how the selection strategies affect performance *if we only use a part of the collected programs* (see Fig 7).
>
> We also think that categorizing and predicting the category of each program is an interesting research direction. We do this in Section 5, where we categorize them based on performance, analyze the perceptual qualities of these categories, and measure the performance when sampling from them (Figure 7). From our experiments and FID results (Fig 5 and LPIPS in Fig 6) we expect that, if grouped by perceptual similarity, shaders sampled from a single category will perform worse than shaders sampled from different categories, as a single category will be less diverse. We are unaware of other categorization strategies that should be applied to make the paper complete, and would appreciate extra feedback from the reviewer regarding this.
>
> FractalDB is not included as part of the dataset (as well as other previous methods). Although there may be some collected programs that resemble Fractals, dead-leaves or other of the previous methods, these constitute a small amount of the training data (as seen in Fig 2 and in the Supp.Mat Figs 3-8). The programs that may resemble fractals and previous methods are uncurated and have not been tuned to perform well on representation learning (as is the case of previous works).
>
> **Correction of reviewers summary**: *Procedural image programs are* **not** *image representation learning methods*. They are programs to procedurally generate images, which we use as data for a representation learning algorithm instead of real images.

---

> ### Author Response · Authors · 2022-08-08
> **For reviewer mJZE**
>
> Could the reviewer engage in the discussion of our rebuttal, specially the following points of their review, which we believe to be incorrect:
> - ***W1. only shows the performance of their approach on the ImageNet dataset*.** We perform experiments on 21 dataset-tasks.
> - ***W3. More experimental results: What happens if we use only a part of the collected programs?*** This particular experiment is studied in detail in Sec 4.1 (see Fig 3) and Sec 5) A collection of shaders.
>
> Although clarity can always be improved, we believe that these two weaknesses raised by the reviewer are especially clear and not **W2. hard to follow** from Tables/Figures (e.g. 21 different columns on Tables of the main paper and Supp.Mat, instead of just 1 column, shows that we perform experiments on 21 datasets).

---

### Author Response · Authors · 2022-07-28
**For all reviewers**

We thank the reviewers for their insightful suggestions and feedback. We appreciate that reviewers have found the contributions of our paper to be very (vdrw) and practically (CTGk) important. We will include their suggestions in the revised version of the paper as explained in the responses to each of the authors.

---

### Meta-Review · Area_Chair_nqQm · 2022-08-31

**Recommendation:** Accept
**Confidence:** Less certain

**Metareview:**

This work presents a method for training neural nets on synthetic data. This data is collected from a collection of thousands of OpenGL programs that rendered images, which are then used for representation learning. The big advantage of this approach is that it avoids of a lot of the biases that are present in natural image datasets. The proposed method is competitive for supervised and unsupervised scenarios. I find the results, especially those with finetuning (as done during the rebuttal period) relatively compelling.

While I agree with reviewer vdrw that, on the whole, the major contribution (an image collection) is not very strong, I still think this kind of approach will be widely interesting to the NeurIPS community. Precisely because the work shows carefully (albeit empirically only) that procedurally generated datasets could be useful for representation learning, especially if you want to avoid the various pitfalls of natural image sets.

**Award:**

No

---

### Decision · Program_Chairs · 2022-09-14

Accept